DOI: 10.1038/s41467-018-07537-8　　**OPEN**

# Opposite macroevolutionary responses to environmental changes in grasses and insects during the Neogene grassland expansion

Gael J. Kergoat [1], Fabien L. Condamine[2], Emmanuel F. A. Toussaint[3], Claire Capdevielle-Dulac[4], Anne-Laure Clamens[1], Jérôme Barbut[5], Paul Z. Goldstein[6] & Bruno Le Ru[4,7]

The rise of Neogene $C_4$ grasslands is one of the most drastic changes recently experienced by the biosphere. A central - and widely debated - hypothesis posits that Neogene grasslands acted as a major adaptive zone for herbivore lineages. We test this hypothesis with a novel model system, the Sesamiina stemborer moths and their associated host-grasses. Using a comparative phylogenetic framework integrating paleoenvironmental proxies we recover a negative correlation between the evolutionary trajectories of insects and plants. Our results show that paleoenvironmental changes generated opposing macroevolutionary dynamics in this insect-plant system and call into question the role of grasslands as a universal adaptive cradle. This study illustrates the importance of implementing environmental proxies in diversification analyses to disentangle the relative impacts of biotic and abiotic drivers of macroevolutionary dynamics.

[1] CBGP, INRA, CIRAD, IRD, Montpellier SupAgro, University of Montpellier, 755 Avenue du campus Agropolis, 34988 Montferrier-sur-Lez, France. [2] CNRS, UMR 5554 Institut des Sciences de l'Evolution de Montpellier, Place Eugène Bataillon, 34095 Montpellier, France. [3] Florida Museum of Natural History, University of Florida, Gainesville, FL 32611, USA. [4] UMR EGCE (Evolution, Génome, Comportement, Ecologie), CNRS-IRD-Univ. Paris-Sud, IDEEV, Université Paris-Saclay, 1 Avenue de la Terrasse, 91198 Gif-sur-Yvette, France. [5] MNHN, Muséum national d'Histoire naturelle (Entomologie), 57 rue Cuvier, 75005 Paris, France. [6] USDA, Systematic Entomology Laboratory, Smithsonian Institution, National Museum of Natural History, PO Box 37012 Washington DC, USA. [7] IRD c/o ICIPE, NSBB Project, PO Box 30772 Nairobi, Kenya. These authors contributed equally: Gael J. Kergoat, Fabien L. Condamine. These authors jointly supervised this work: Paul Z. Goldstein, Bruno Le Ru. Correspondence and requests for materials should be addressed to G.J.K. (email: gael.kergoat@inra.fr)

The emergence and spread of grasslands during the Miocene and Pliocene is one of the most drastic changes recently experienced by the biosphere[1]. Low latitudes and altitudes grassland ecosystems are dominated by grasses (Poaceae) that employ $C_4$ photosynthetic pathways[1,2], and are thereby better adapted to low $CO_2$ conditions than plants using a $C_3$ pathway, especially under warm and dry conditions[3]. The multiple independent evolutionary origins of $C_4$ pathways has led to suggestions that the major drop in atmospheric $CO_2$ beginning roughly 30 million years ago (Ma) provided conditions conducive to the diversification of $C_4$ grasses and their rise to dominance in many areas at the expense of $C_3$ lineages[1,2,4]. Fossil phytoliths and paleosols support this hypothesis, indicating that $C_4$ grasses appeared about 20–25 Ma in the geological record[5]. This pattern is also confirmed by recent molecular phylogenetic studies, which indicate that distinct $C_4$ grass lineages appeared multiple times between the late Oligocene and the late Miocene[1], and that the shift from $C_3$ to $C_4$ pathways constituted an important precursor to speciation in the evolutionary history of grasses[6]. However, it was not until the late Miocene and Pliocene (ca. 3–9 Ma) that $C_4$ grasslands became a major component of Earth's terrestrial ecosystems across all continents[5,7,8]. The drop in atmospheric $CO_2$ was likely instrumental in this transition, but other environmental factors, such as the increased intensity of fire regimes, herbivory, and changes in climatic seasonality are likely causal factors[8–10].

This major biome shift is thought to have had a pivotal role in the evolution of herbivore communities by promoting the diversification of herbivorous lineages associated with $C_4$ grasses[11–13]. Feeding on $C_4$ grasses requires-specific adaptations for the feeders, as tissues from these plants are usually more fibrous than those of $C_3$ plants, due to the higher density of bundles of fibers, veins, and silica phytoliths (reviewed in ref. [14]). They also have lower nutritional values than $C_3$ plants because of lower nitrogen and higher carbon contents[15–17]. In mammals, a commonly accepted theory posits that the Neogene grasslands enabled the development of adaptive radiations, favouring lineages of grass eaters following key innovations in dental morphology (hypsodonty), and plant assimilation[18]. However, this view was recently challenged by studies revealing that hypsodonty originated well before the appearance and spread of Neogene grasslands (e.g. ref. [19]). Toljagic et al.[20] also demonstrated that the evolution of hypsodonty in ruminants was slower than previously thought, excluding the possibility of its representing an immediate adaptive response. Another study revealed that the hypothesized adaptive radiation of horses coinciding with the spread of grasslands was not correlated to phenotypic evolution and, if anything, impeded by ecological limits[21]. In herbivorous insects, the role of $C_4$ grasslands as a potential adaptive zone has never been properly investigated even though several independent species-rich lineages are intimately associated with $C_4$ grasses, exhibiting-specific morphological adaptations, such as more heavily chitinized mandibles (e.g., in acridid grasshoppers[22]), or enlargement of mandibular muscles to cope with silica-rich stems (e.g., in stemborer moths[23]). Dated phylogenies also provide conflicting evidence, suggesting that diversification of specialized insect lineages either predated (e.g., for satyrine butterflies[24]), or coincided with (e.g., in anisopliine beetles[25]) the spread of $C_4$ grasslands. No study to date has quantified whether reciprocal macroevolutionary dynamics occurred between $C_4$ grasses and specialized lineages of herbivores. Under the theory of ecological opportunity[26], we can hypothesize that $C_4$ grasslands provided a template for the radiation of specialist herbivores able to colonize this new niche. Alternatively, we could also postulate that the diversification of specialized herbivores may also be unaffected by host diversification, if they remain able to utilize all the descendant hosts of their ancestral host-plants.

To investigate the diversification dynamics of both herbivorous insects and grassland plants in the context of the $C_3/C_4$ transition, we study a clade of stemborer moths (Noctuidae: Apameini: Sesamiina) representing a radiation of ca. 200 species, most of which are distributed in their Afrotropical centre of origin[27]. Sesamiina are mostly associated with $C_4$ grasses[28–30] and are well-adapted to Afrotropical grasslands, where they constitute one of the most abundant and conspicuous moth group[28]. Similar to other groups of internal feeders[31–35], Sesamiina species are highly specialized; most species only feed on one or a few plant species. In any given locality diverse sesamiine species are also usually associated with distinct sets of host-plants. They constitute a particularly tractable system for studying co-diversification given the expanding body of accurate life history data surrounding host-plant associations.

We explore the macroevolution of Sesamiina stemborers in the light of comprehensive phylogenetic, ecological, and geographic data for both the herbivorous insect group and their host-plants (Supplementary Data 1–3). To determine whether the origin of Sesamiina and their $C_4$ grass hosts coincided, we conduct phylogenetic and dating analyses on a novel molecular dataset of six loci, comprising 1393 specimens from 245 noctuid species, and historical biogeographic analyses based on a timeframe established for Poaceae[6]. We also carry out several estimations of ancestral character states to investigate the evolution of host associations, and ecological attributes. To complement these approaches, we use an array of diversification analyses, including a recently developed analytical method[36] to test the potential role of several paleoenvironmental proxies within a temporal framework for both insects and plants. As we describe, the double-dating of plant and insect clades shows synchronous origins and radiations, suggesting that they interacted and diversified in parallel. We find strong evidence that, following the origin of both clades, speciation rates of insects decreased, whereas those of grasses increased through time. Our results also reveal opposite macroevolutionary responses to environmental changes with temperature as a primary evolutionary driver. This study confirms that the drop of atmospheric carbon is a key driver for the diversification of $C_4$ grasses. Analyses using organic carbon concentration as a proxy for the proportion of $C_4$ grasses also indicate that the expansion of $C_4$ grasslands did not act as a primary environmental driver for the insects. This study therefore suggests that the rise of $C_4$ grasslands did not necessarily favor the diversification of herbivores that feed on them.

## Results

**Synchronous temporal and spatial diversification.** We assembled a dataset for a group of $C_4$ grass specialist moths with information on geographic distribution, host- and habitat-associations (see Methods). A total of 1393 specimens representing 245 noctuid species (including 181 species of Sesamiina) were sampled for six loci (Supplementary Data 1); this sampling encompasses all described Sesamiina genera and ca. 90% of the described Sesamiina species. Phylogenetic analyses were carried out in maximum likelihood and Bayesian frameworks, using all individuals (specimen-level dataset) and using only one representative per species (species-level dataset). The resulting trees (Supplementary Figures 1–2) provide a comprehensive and robust picture of Sesamiina relationships (within Sesamiina ca. 80% of nodes are well supported for the species-level dataset). Molecular dating analyses performed on the species-level dataset with Bayesian relaxed clocks using uniform priors and either birth-death (BD) or pure birth tree priors yielded consistent

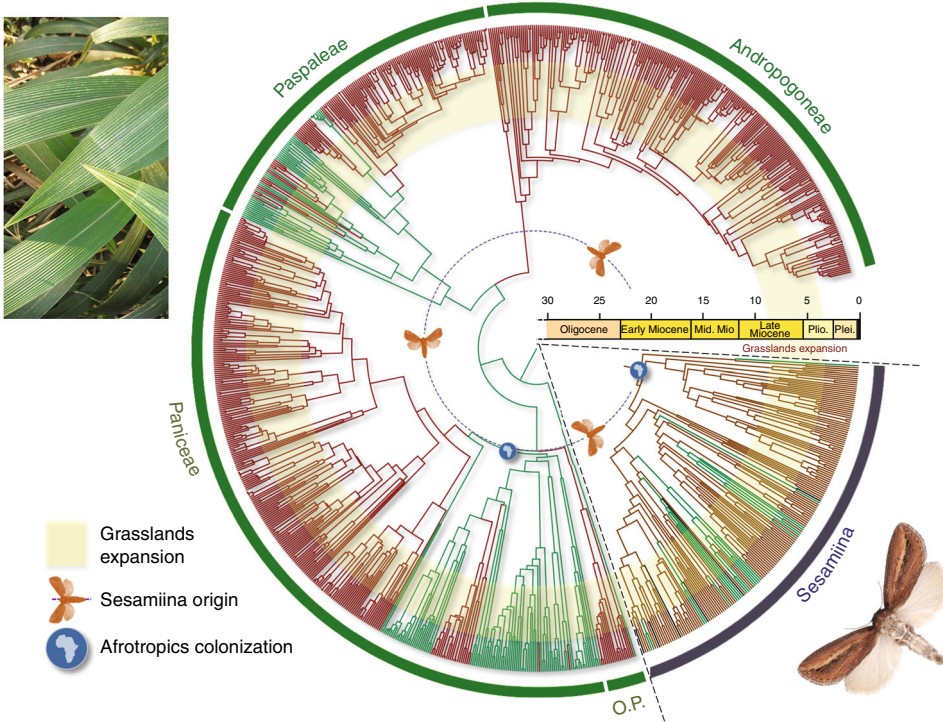

**Fig. 1** Double-dating of associated stemborer moths and grass lineages. Dating analyses of panicoid grasses and stemborer moths reveal synchronous origins and radiations between Afrotropical $C_4$ lineages and their herbivorous moths. Ancestral state estimations show the evolution of photosynthetic pathway for grasses and the evolution of host-type preferences for moths ($C_3$ in green $C_4$ in brown). For illustrative purpose, a picture of *Setaria megaphylla* (Steud.) T.Durand & Schinz is presented on the left, and a picture of *Acrapex subalbissima* Berio is figured on the bottom right. Copyright notes: the picture of grass was cropped from "Leaves of Setaria megaphylla at Umdoni Bird Sanctuary, Amanzimtoti, South Africa" by M. Purves, licensed under CC BY-SA 3.0. The picture of the moth was taken by B. Le Ru (last author of the paper)

divergence time estimates for the insect clade (Supplementary Figures 3–4), with the BD model being recovered as the best tree prior ($MLE_{Yule} = -84,860.12$ vs. $MLE_{BD} = -84,002.91$, Bayes factor » 10). For all downstream analyses, we thus relied on the chronogram obtained with the BD tree prior. Based on this dating analysis, Sesamiina originated ca. 20.7 Ma (95% credibility interval = 17.8–23.2 Ma), and ancestral state estimation (ASE) revealed a high level of phylogenetic niche conservatism (PNC) on $C_4$ lineages of the grass subfamily Panicoideae (Supplementary Figures 5a–b, 6a–b), a diverse group of plants (about 3500 known species) with a worldwide distribution[37,38]. Sesamiina are mostly associated with the $C_4$ tribe Andropogoneae (1132 species worldwide[6]) and a large $C_4$ clade (732 species worldwide[6]) of the tribe Paniceae. These two panicoid clades represent almost two-thirds of the diversity of the subfamily (Sesamiina are rarely associated with the tribe Paspaleae, which is mostly distributed in the Western Hemisphere; Supplementary Data 3). Although the inferred pattern of PNC is conserved (few host-shifts are inferred), ASE analyses indicate that at least 19 nodes represent a generalist state to at least one descendant specialist (Supplementary Figure 5a). These results are particularly relevant in the context of the current debate on the evolution of host-range of phytophagous insects[39–41], surrounding two hypotheses with different points of emphasis: the oscillation hypothesis (OH), in which speciation is driven by expansions and contractions of host-ranges (usually considered at the plant family level[39–41]), and the musical chairs hypothesis (MCH), where speciation is driven by host-plant shifts, without concomitant changes in host-ranges or diet breadth. Both hypotheses are similar in that they predict more species in clades with more 'lability' in host use (Nylin pers. com.). The elevated occurrence of transitions

between generalist to specialist states is more consistent with the OH, which predicts that 'host expansions do not happen with regular frequency and amplitude'[41] relative to host breadth contractions, and that clades with such oscillation patterns should be more diverse[39,41] than their counterparts. Also consistent with the OH is the observation that: (i) generalist Sesamiina species (such as *Sesamia calamistis* or *S. nonagrioides*) tend to be widespread, and (ii) recolonizations of ancestral hosts that have been lost (such as Panicoideae) are common (Supplementary Figure 5a). An increase in speciation rates following inferred shifts in diet breadth was not observed within the Sesamiina clade (see Supplementary Table 1). However, this does not necessarily represent an invalidation of the OH, as the increase of speciation rates linked to host-range oscillations may not be detectable at this macroevolutionary scale; its assessment likely requires denser sampling throughout the stemborer subtribes (especially within the diverse Apameina, which are primarily associated with $C_3$ plants in the Northern hemisphere[23]) and their relatives.

To get a better understanding of the spatial patterns of the primary sesamiine host-plant group, the Panicoideae, we reconstructed its biogeographic history and infer that the first colonization of the Afrotropics by a panicoid lineage (tribe Paniceae) occurred ca. 21 Ma, slightly before the start of the Sesamiina diversification on the same continent ca. 20.7 Ma (Fig. 1 and Supplementary Figure 7). The early diversification of panicoid lineages in the Afrotropics is mostly attributable to the Paniceae, as Andropogoneae only colonized the Afrotropics in the late Miocene (Supplementary Figure 7). We acknowledge that the origins of Poaceae and internal clades are debated due to a lack of consensus on the use of fossil calibrations. Two main strategies are used here to estimate divergence times of grasses:

the first relies on macrofossils that are reliably assigned to a given clade, whereas the second also includes microfossils (phytoliths) whose use in molecular dating is controversial because phytolith characters are usually phylogenetically unreliable[42,43]. Accordingly, the grass divergence times can vary depending on the calibration strategy. Here we rely on the results of the calibration strategy with macrofossils of the comprehensive study of Spriggs et al.[6], which recovered an origin for Paniceae at ca. 21 Ma. Though older age estimates were sometimes inferred from the use of phytoliths[6,43], these are questionable because age estimates for $C_4$ grass lineages then significantly predate the fossil record of open grasslands. The ca. 21 Ma age estimate for the Paniceae is also consistent with the results of the macrofossil calibration of Christin et al.[43] (median age of 25.4 Ma with a 95% credibility interval = 20.6–31.7 Ma) and with the results of the microfossil calibration of Prasad et al.[42] (median age of 20.1 Ma with a 95% credibility interval = 11.5–29.6 Ma). Importantly, it is also in accordance with our inferred origin of stemborer moths (95% credibility interval = 17.8–23.2 Ma). Overall, these results thus support the nearly simultaneous and sympatric origin of both Sesamiina and panicoid grasses, and their co-diversification in the Afrotropics during the early Miocene.

**Diversification rates of moths and grasses are uncorrelated.** Having established that Sesamiina and panicoid grasses colonized the Afrotropics roughly at the same time, we investigated whether the diversification dynamic of moths paralleled that of the panicoid grasses. We hypothesized that the spread of $C_4$ grasslands in the Neogene promoted the diversification of moths by providing a newly available resource and ecological opportunity for these insects already associated with Poaceae. To test this hypothesis, we used three independent approaches to estimate the diversification dynamics in each clade. First, we used Bayesian analysis of macroevolutionary mixtures (BAMM[44]), which found strong support for a single diversification regime showing elevated speciation rates in the early stages of the insect lineage evolution, followed by a progressive slowdown but no drastic downward or upward shift in speciation (Supplementary Table 1 and Supplementary Figure 8a). In contrast, we found evidence for the opposite pattern in panicoid grasses, with lower rates close to the origin of the group and increasing rates of speciation over time, accompanied by two significantly supported upshifts in speciation with the crown tribes Andropogoneae and Paspaleae (Supplementary Table 1 and Supplementary Figure 8b). Because of the potential sensitivity of BAMM[45,46], we used two other approaches to test the pattern of inverted diversification rates. Analyses inferring speciation rates as a function of time (RPANDA[47]) recovered distinct diversification patterns, with decelerating speciation rates for the moths and accelerating speciation rates for the grasses (Fig. 2a). Using episodic BD models (TreePar[48]), we also recovered two significant shifts in diversification at 13.0 and 1.3 Ma for the moths, both consistent with progressively decelerating diversification (Fig. 2b and Supplementary Table 2). For the panicoid grasses, episodic BD models inferred three shifts of diversification at 20.3, 9.6, and 1.5 Ma but, again contrary to the insect clade, these shifts represented increased diversification (Fig. 2b and Supplementary Table 2b). The congruence of these results suggests that the diversification rates of both groups are inversely correlated. Analyses relying on the binary-state speciation and extinction (BiSSE) model[49] recovered very similar net diversification rates for both Afrotropical and non-Afrotropical panicoid lineages (comprised between 0.24 and 0.28 events per Myr per lineage, respectively). Lineages-through-time (LTT) plots show patterns of increasing species accumulation for both Afrotropical and non-Afrotropical

panicoid lineages, with no distinct up-shifts and no plateau of diversity (Supplementary Figure 9a–b). By contrast the stemborer LTT plot exhibits a plateau of diversity toward the present (Supplementary Figure 9c). Overall, the extent of the analyses performed suggests that diversification is similar between Afrotropical and non-Afrotropical panicoid lineages; however, we acknowledge the uncertainty underlying this finding because of the incompleteness of the sampling for the panicoid grasses (805 species out of ca. 3500 known species[6]).

**Differential responses to environmental changes.** We used a maximum-likelihood framework to test competing hypotheses of rate variation based on BD models. Out of 14 diversification models, a model with temperature-dependent speciation best fit the data for both groups (Supplementary Table 3). Best-fit models further indicate a positive correlation ($\alpha > 0$) between temperature and speciation for the Sesamiina while they show a negative correlation ($\alpha < 0$) for the panicoid grasses, indicating that the groups had opposite responses to changes of temperature over time. For Sesamiina, speciation rates appear to have been higher under warmer climates while the opposite holds for panicoid grasses (Fig. 2c, d). In Sesamiina, diversification models relying on $\delta^{13}C_{Organic}$ (as estimated from stable carbon isotopes; see Methods) as a proxy for the proportion of $C_4$ grasses show a negative correlation (Fig. 2e, f), and are consistently outperformed by diversification models incorporating temperature or time dependence (Supplementary Table 3). The latter and the fact that no upshift in diversification rates was inferred for the Sesamiina during the spread of $C_4$ grasslands suggest that the expansion of $C_4$ grasslands did not act as a primary environmental driver of sesamiine diversification. To confirm this pattern, we performed additional diversification analyses using the recently estimated proportion of $C_4$ plants in Eastern Africa through time[50] instead of the global $\delta^{13}C_{Organic}$. We still found that a positive temperature-dependent model best explains the stemborer diversification, and we found a negative $C_4$ grasses-dependent model suggesting a macroevolutionary decorrelation between Afrotropical $C_4$ plants and their herbivores (Supplementary Table 4). In panicoid grasses as a whole (i.e., both $C_3$ and $C_4$ lineages), models accounting for changes in atmospheric $CO_2$ also ranked behind models incorporating temperature or time dependence. However, when independently analyzing the two primary panicoid $C_4$ host-plant lineages (i.e., Andropogoneae and the large clade of Paniceae using a $C_4$ pathway), models accounting for changes in atmospheric $CO_2$ is recovered as best-fit models (Supplementary Table 3). This is consistent with the predictions made by Pagani et al.[51] and Edwards et al.[1], who postulated that the diversification of $C_4$ grass lineages is correlated positively with the drop in atmospheric $CO_2$ (Fig. 2e, f). It is important to note that the pattern of increasing speciation rates is always recovered for the primary host-plant groups, independently of the clade being considered (panicoid grasses as a whole, Andropogoneae or the major clade of Paniceae).

The contrasting patterns recovered for Sesamiina and panicoid grasses may best be explained by their long-term responses to environmental changes, and in particular to temperature. Grass diversification ($C_3$ and $C_4$ lineages) correlated positively with climatic cooling and, in the main $C_4$ lineages, with the drop in atmospheric $CO_2$. By contrast, lower temperatures appear to have slowed diversification of the insects. Interestingly, changes in temperature over time (Fig. 2d) are also associated with changes in climatic conditions: during the Cenozoic, warmer periods have tended to be associated with wetter conditions, and colder periods are associated with drier conditions, with a shift from moist woodlands/wetlands and predominantly summer rainfall to a

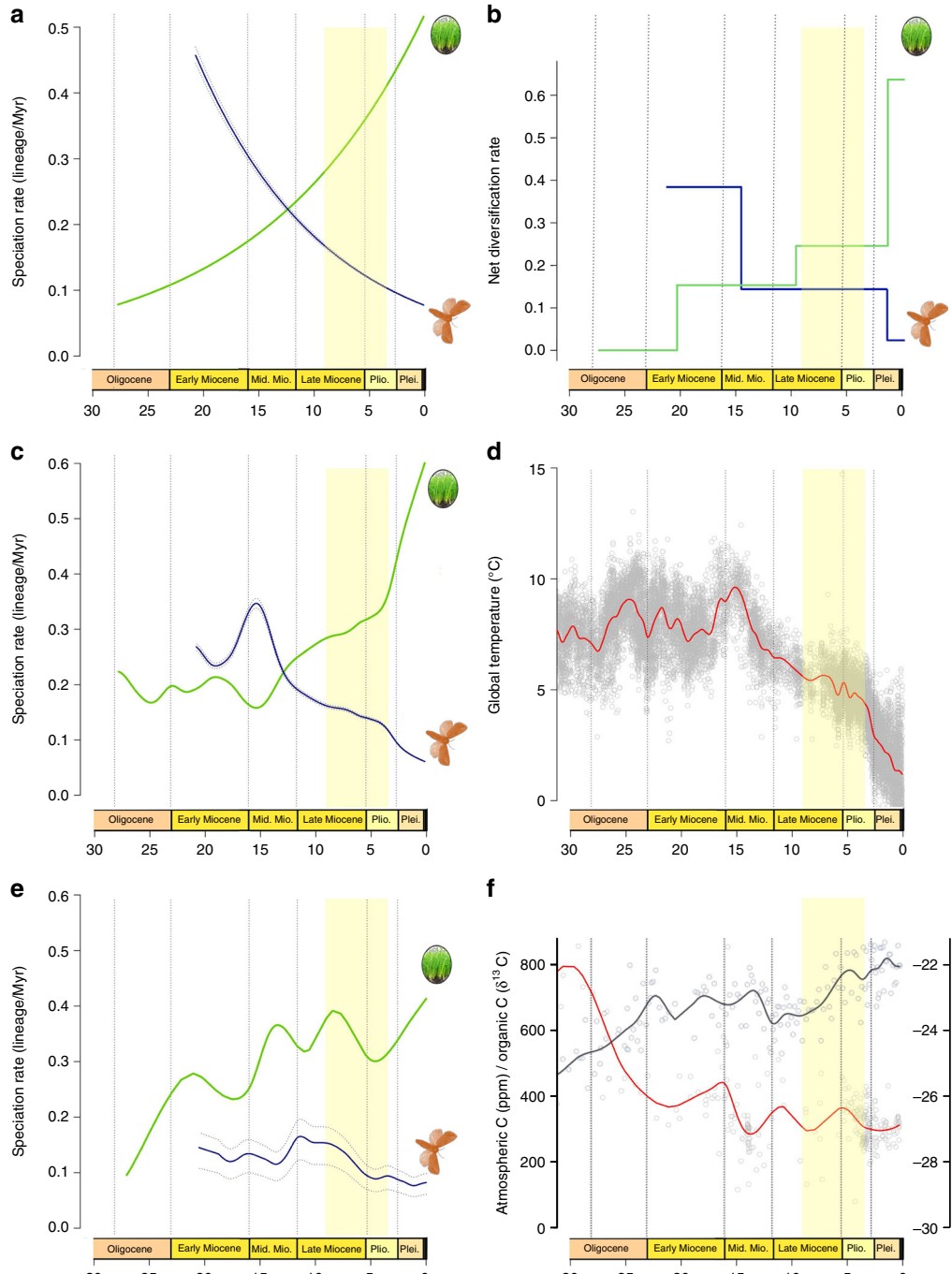

**Fig. 2** Diversification dynamics through time and potential environmental factors. **a** Diversification dynamics through time for the stemborer moths (blue curve) and the panicoid grasses (green curve) inferred with a time-continuous BD model. **b** Diversification dynamics through time with an episodic BD model. Both analyses show an opposite trend of speciation rate evolution for moths and grasses. **c** Analyses of temperature-dependent diversification indicate that past warm climate fostered moth speciation, whereas cooling events boosted grass speciation. **d** Global temperature inferred from $\delta^{18}O$ isotopes in benthic foraminifer shells recovered in marine sediments[41]. **e** Analyses linking speciation and past variations in atmospheric $CO_2$ for grasses, suggest higher speciation when $CO_2$ concentration decreases, although the analyses linking for moths show a negative association between speciation and the rise of grasslands. **f** Potential environmental factors driving the diversification of stemborer moths and panicoid grasses such as the increase of $C_4$ grasses (dark blue curve) reconstructed from $\delta^{13}C$ of tooth enamel of mammalian herbivores[52, 53], the atmospheric concentration of $CO_2$ (red curve) compiled from multiple sources (notably inferred from $\delta^{13}C$ measurements[52]). Copyright notes: the grass and plant symbols rely on pictures taken by G.J. Kergoat and B. Le Ru (first and last authors of the study, respectively)

range of more arid to semi-arid open habitats and increasingly seasonal rainfall regimes[52–54]. From a biological point of view $C_4$ grasses are less drought sensitive than $C_3$ grasses[55], and generally benefit from drier conditions associated with high-light habitats[1].

In Sesamiina, numerous species are hygrophilous and are distributed in grasslands in contact with wet habitats (e.g., see refs. [29,30] Supplementary Data 2). To investigate whether Sesamiina exhibit a pattern of phylogenetic niche conservatism

for a given climatic condition, ASE of moths' ecological preferences were performed. Results (Supplementary Figure 10) suggest that wet habitat-association is conserved phylogenetically. This leads to the hypothesis that the decrease of wet habitats and the increase of periodically dry conditions associated with progressive temperature drops impeded the diversification of Sesamiina over time.

**Role of $C_4$ grasslands in herbivore speciation.** Our results also challenge the role of $C_4$ grasslands as a potential adaptive zone, and as a driver for herbivorous insect speciation. The spread of $C_4$ grasslands is usually seen as an ecological opportunity, providing a fertile ground for speciation through adaptation and diversifying selection. Ecological opportunities depend on the availability of novel ecological resources coupled with ease of colonization and/or adaptation or key innovation. Here we argue that the radiation of Sesamiina was likely influenced by two-related ecological opportunities: (i) access to a novel resource—the $C_4$ grasses—in the Afrotropics in the early Miocene, and (ii) a major biome shift that enabled the spread of such $C_4$ grasslands and reflected in the rapid radiation of $C_4$ grasses themselves in the late Miocene. We found that these parallel and extended events impacted insect diversification in different and far less straightforward ways than previously believed.

Our results indicate that the diversification of sesamiine stemborers is tightly linked to the emergence and diversification of $C_4$ grasses at the time when both interacting partners colonized the Afrotropics in the early Miocene. This new ecological resource provided some of the requisite conditions for the radiation of specialist insects. As diversification proceeded, we infer that the magnitude of this novel resource diminished as the window of ecological opportunity closed with the saturation of niche space by new species. Therefore, a core prediction of this scenario is that speciating lineages would undergo early bursts of rapid diversification followed by asymptotic decreases in diversification rates over time[56]. Here all diversification analyses recover a signal consistent with an early burst pattern of diversification among ancestral Sesamiina, which is thus consistent with a hypothesis of an adaptive radiation in the broad sense. Eventually speciation rates are expected to slow towards an equilibrium as species saturate available niches (in this case, $C_4$ grass species). To test this notion, a diversity-dependent model of diversification[57] was used to assess whether the entire Sesamiina clade approached stasis, interpreted as an equilibrium or evolutionary carrying capacity of sorts. In this case, stemborer radiation conforms to a diversity-dependent speciation model (Supplementary Table 5) that predicts a carrying capacity of 240 species (ca. 200 species are known today). This suggests the existence of ecological, and in this case spatio-temporal, limits to moth diversification. Such a scenario is traditionally interpreted as adaptive radiation through the occupation of extrinsic niches or niche filling[56].

Yet, niche space of sesamiine stemborers presumably expanded over time both in terms of diversity and abundance due to increased diversification rates of $C_4$ grasses over time, and their rise to dominance starting in the late Miocene (green curve in Fig. 2a–d, see also Supplementary Figure 9a–b and Supplementary Methods for evidence showing similar $C_4$ species accumulation in Afrotropics and outside)[1,6]. This increase in niche space therefore constituted a major ecological opportunity for $C_4$-specialist insects. However, instead of the expected upshift of speciation/diversification rates, a decline in the diversification of moths was inferred (blue curves in Fig. 2). The pattern of diversification between moths and $C_4$ grass lineages appears decoupled, as indicated by the fact that models with $\delta^{13}C_{organic}$

dependence (our proxy for the proportion of $C_4$ grasses) are negatively correlated. To explain this counter-intuitive pattern, we postulate that inverse responses to abiotic factors account for differences in the pace of diversification. When the climate changed drastically in the late Miocene, becoming cooler and drier, $C_4$ grasses began an unprecedented radiation that did not confer higher speciation rates to their moth herbivores. In fact, as the climate cooled, sesamiine moths diversified less rapidly despite the fact that their host-plants were thriving.

The marked preference of Sesamiina for wet habitats may have limited the dimensions of their available niche space in an increasingly dry environment. Dry environments also directly influence plant form and phytolith formation in $C_4$ grasses. As underlined by a recent large-scale study on West African grasses[58], drier environments are characterized by smaller, more therophytic grass species, which are too small to accommodate all stemborer larval stages. Sesamiina larvae are also sensitive to the high levels of silicon content[59]. One of the factors promoting phytolith formation (accumulation of silicon) in grasses is evapotranspiration[60], which is more important in drier environments[61]. All of these observations are consistent with a reduction of the availability of suitable niches in dry environments and may reflect or even explain the preference of Sesamiina for wet habitats. An interesting parallel may be drawn here between stemborers and another group of $C_4$-specialists, the grass-feeding *Bicyclus* butterflies (Nymphalidae, Satyrini), which represent a radiation of ca. 103 species in the Afrotropics[62]. Nokelainen et al.[14] discovered that, when laying eggs, females of *Bicyclus safitza* (Westwood) preferred shaded habitats characterized by high concentrations of tender grass foliage to open habitats dominated by less readily consumable leaves. We surmise that the advent of drier and more open environments in the late Miocene may have limited the diversification of Sesamiina. Despite representing a highly apparent and diverse food source, the confinement of these grasslands to open and comparatively dry habitats translated to a reduction of potential niche space for these grass-specialist insects.

## Discussion

Our study shows that differential responses to environmental changes may yield counter-intuitive patterns of diversification in plants and insects. Although the initial radiation of $C_4$ grasses in the Afrotropics likely met criteria for an adaptive zone in a group of grass-specialist insects, subsequent climatic changes may have driven diversification in a direction opposite that predicted by the inferred spread of $C_4$ grasslands ca. 3–9 Ma. Environment-dependent diversification analyses demonstrate that the diversification of at least two major $C_4$ panicoid grass lineages comprising the primary hosts of Sesamiina was likewise correlated with a drop in atmospheric $CO_2$. The negative correlation between the spread of grasslands calibrated via $\delta^{13}C_{organic}$ proxy and the diversification of Sesamiina further calls into question the role of $C_4$ grasslands as simple drivers of graminivorous herbivores. This study suggests a potentially far more complex picture regarding the role of grasslands as herbivore adaptive zones analogous to the results of recent studies on mammals (e.g. ref. [21]). We also illustrate the potential of implementing multiple environmental proxies in diversification analyses that serve to disentangle the relative impacts of biotic and abiotic drivers of macroevolutionary dynamics.

## Methods

**Moth taxon sampling and molecular dataset.** Extensive field surveys conducted since 2004[28] in 17 sub-Saharan countries and in the Palearctic region (in France and Portugal), targeting wild habitats rich in Poales combining infested host-plant collections and light traps, allowed us to obtain about 80,000 specimens, of which

more than 52,000+ larvae were reared (Supplementary Figure 11) from 220 identified species of grasses and sedges. A total of 1389 specimens from 241 noctuid species were sequenced; we also used sequences from GenBank for four additional species (Supplementary Data 1). This sampling encompasses all described Sesamiina genera and ca. 90% of the described species. We sequenced 2.6 kilobases of the mitochondrial genes *cytochrome c oxidase I*, *cytochrome b*, *ribosomal 12S*, and *ribosomal 16S*, and 2.1 kilobases of the nuclear protein-coding gene *elongation factor-1α* and the *ribosomal 28S*. The individual gene fragments were aligned and the reading frames of protein-coding genes checked.

**Moth phylogenetics and dating**. Partitioning schemes and models of nucleotide substitution were estimated. Phylogenetic relationships for the specimen-level dataset were inferred with maximum likelihood using RAxML[63]. For the species-level dataset (245 species), we reconstructed phylogenetic relationships with Bayesian inference using MrBayes[64], but also with RAxML and IQ-TREE[65] for maximum likelihood to estimate branch support in all analyses. Resulting phylogenetic trees were used to estimate divergence times with a Bayesian relaxed-clock approach in BEAST[66], with one clock model for the mitochondrial genes and one clock model for the nuclear genes (see Supplementary Methods). Given the scarcity and questionable status of noctuid fossils, we relied on three secondary calibrations (see Supplementary Methods). We used uniform priors to model conservative node calibrations.

**Ancestral host-plant associations and ecological preferences**. Host-plant associations and ecological preferences (Supplementary Data 2) were determined by field surveys and rearing of larvae from plants collected in the field (see Supplementary Note 1). Host associations were either categorized at the family level (for plants belonging to Cyperaceae and Typhaceae) or at the subfamily level (for plants belonging to Poaceae). We were unable to assign a host-plant state for seven of the 181 species, because only adults were collected; therefore, we coded them as ambiguous. We used multistate ASE to account for the few species that develop on distinct families/subfamilies (10 species associated with distinct plant families; eight species associated with distinct grass subfamilies). Ancestral character states were reconstructed with a maximum likelihood approach as implemented in the DEC model[67]. We also carried out an analysis of a truncated dataset by pruning all Sesamiina species without genus- or species-specific host-plant data. To avoid favoring the OH over the MCH in the ASE analyses, we used an unconstrained transition matrix allowing any host shift to be equally probable. We also conducted two supplementary analyses with binary traits (see Supplementary Data 2 for the coding). First, we explored the evolution of host-plant preference in relation to photosynthetic pathways ($C_3$ or $C_4$). Second, we inferred the evolution of ecological preferences in relation to dry or wet environments.

**Plant phylogeny and biogeography**. We studied the origin and diversification of the main host-plant group. As we inferred that Panicoideae was the ancestral state of host-plant preference for the stemborers, we obtained a dated phylogeny for 805 species of Panicoideae by pruning the dated Poaceae tree of Spriggs et al.[6]. We tested the hypothesis of a synchronous temporal and geographic origin between the Panicoideae and their insect predators (knowing that the radiation of stemborers originated in the Afrotropics[27]). Ancestral area estimations for Panicoideae were inferred with the DEC model. Species ranges were coded by presence–absence (excluding marginal distribution or anthropic introduction) for seven areas (Afrotropics, Australasia, Eastern Palearctic, Nearctic, Neotropics, Oriental, and Western Palearctic; Supplementary Data 3). The historical biogeography of Panicoideae was estimated with an unconstrained DEC model with only one time slice (see Supplementary Methods for details).

**Diversification of insects and plants**. We investigated whether the modes of species diversification were similar between the Panicoideae and the insects feeding on them, following a step-by-step procedure that took time-calibrated phylogenies as the basis for the analyses. When possible we took into account age uncertainties (100 randomly selected trees from the posterior distribution of dating analyses). We first estimated whether the clades' diversification deviates from a constant BD model and if shifts in speciation and extinction rates occurred using three complementary approaches (BAMM[44]; RPANDA[47]; and TreePar[48]). The best-fit model was selected using Bayes factors (BAMM), corrected Akaike information criteria (RPANDA) and likelihood ratio tests (TreePar). We further tested the impact of past environmental changes on the diversification of both clades to assess whether they responded similarly to changing abiotic conditions. An environmental-dependent model was used in which speciation and/or extinction can vary as a function of temporal variation of the environment[36]. We compiled paleodata (Fig. 2e, f) for variables that have been associated with macroevolutionary hypotheses of diversification and that represent best paleoclimatic changes: temperature data (inferred from $\delta^{18}O$ measurements[54]) and atmospheric $CO_2$[68]. Temperature is the canonical indicator of climate change. Levels of atmospheric $CO_2$ are thought to impact photosynthetic organisms[2,3,8]. We also obtained $\delta^{13}C_{organic}$ reflecting global changes in organic carbon sequestration[69,70]. In the Cenozoic, $\delta^{13}C$ is the best proxy for reconstructing ancient $C_4$ grasslands as it

represents the proportion of $C_3$ and $C_4$ grasses[1,2,50,70], and is therefore relevant to test the role of grasslands on the evolution of herbivores that feed on grasses. Recently, a compilation of $\delta^{13}C$ for eastern Africa has allowed quantification of the percentage of $C_4$ plants through time[50]. The percentage of $C_4$ through time provides a more direct test of the role of grasslands. For stemborers we fitted diversification models in which speciation and/or extinction may vary according to dependence with temperature and $\delta^{13}C_{organic}$/percentage of $C_4$ in eastern Africa. For Panicoideae grasses we fitted models with temperature- and $CO_2$-dependence. These models were compared to constant-rate and time-variable models. Finally, we also used the BiSSE model[49] to estimate whether the respective diversification rates of Afrotropical and non-Afrotropical panicoid species are comparable. To do so, first we estimated the number of native Afrotropical panicoid species to be comprised between 900 and (conservative estimate) and 1034 species (see Supplementary Methods for details). We then carried out eight BiSSE models, from the null model (no variation of rates) to the most complex model in which all parameters of speciation, extinction and transition are estimated (see Supplementary Methods for details). We also generated LTT plots for the panicoid grasses, based on a 805 species timetree; distinct plots for both Afrotropical and non-Afrotropical lineages relied on distinct BiSSE analyses, with an estimated diversity of 900 and 1034 species for the Afrotropical lineages. A LTT plot was also generated for the Sesamiina, based on a 181 species timetree.

## Data availability
All newly generated sequencing data that support the findings of this study have been deposited in GenBank (accession numbers MH847792-MH853351). All other relevant data are available as Supplementary Data.

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

## Acknowledgements

We would to thank A. Kawahara and P. Arnal for many helpful comments. Financial support was provided by IRD and icipe, African Insect Science for Food and Health, and by the project IMPACT_PHYTO funded by the INRA. Part of the sequencing was also supported by the programme BdV (Project NSBB) supported by a joint CNRS, INRA, and MNHN consortium. Laboratory facilities were provided by CBGP and EGCE in France and icipe in Kenya. We want to thank the numerous people who helped with fieldwork, rearing of larvae, collection permits, or studies of museum material (see Supplementary Note 2 for a detailed list). All specimens were collected under appropriate collection permits and no conflicts of interest were discovered. Mention of trade names or commercial products in this publication is solely for the purpose of providing-specific information and does not imply recommendation or endorsement by the USDA; USDA is an equal opportunity provider and employer.

## Author contributions

G.J.K., F.L.C., E.F.A.T., P.Z.G., and B.L.R. designed research; J.B. and B.L.R. sampled and identified the specimens; E.F.A.T., C.C.-D., and A.-L.C. generated the molecular data; G.

J.K. and F.L.C. analysed the data; and G.J.K., F.L.C., E.F.A.T., P.Z.G., and B.L.R. wrote the paper.

## Additional information

**Competing interests:** The authors declare no competing interests.

