## [Peer Review File · Nature Communications]

Reviewers' Comments:

Reviewer #1:

Remarks to the Author:

This paper explores the hypothesis that the global expansion of C4 grasslands during the Miocene allowed the diversification of C4 grass herbivores. As a study system, it uses the Sesamiina, a group of moths feeding on C4 plants, and especially panicoid grasses of the subtribe Andropogoneae (the group of grasses that dominates tropical grasslands). Using phylogenies for the host and herbivores, it shows that both groups emerged at the same time around 20 Ma. However, while the diversification rate of Andropogoneae increased through time, that of Sesamiina decreased through time, leading to an apparent inverse correlation. Palaeoclimate models then identify different drivers of diversification in the two groups, with temperature correlating positively with speciation in the insects, but negatively in the Andropogoneae grasses. The authors conclude that different factors dictate the diversification rates of different groups, so that grassland expansion did not universally lead to the diversification of the associated herbivores.

This paper is analytically solid, and explores an important issue; the drivers of diversification. It can therefore make a strong contribution to the literature, although there are some conceptual issues that I think require clarification, through additional analyses and/or clarification in the text. I list these below.

GENERAL POINTS:

1) The authors focus on the Andropogoneae grasses, although other groups of grasses and sedges are consumed by the moths. Since the Andropogoneae are the main target of the insects, this can be justified. However, if I understood well, the analyses consider the total diversity of Andropogoneae without consideration for the proportion of these that can serve as hosts for the insects. I could imagine a situation where only a fraction of the Andropogoneae can feed the insect, so that the total diversity of the group is a poor indicator of the niches available for the moths. While these can include species-specific properties, an obvious condition is the geographic distribution. Since the moths are distributed in Africa, the diversity of Andropogoneae in Africa is the only thing that matters, but the diversification analyses consider the overall Andropogoneae distribution. Can the authors verify that the African diversity of Andropogoneae increased at the same pace as the overall diversity of this widely distributed group?

2) The authors hypothesize that the diversity of grasses would increase the diversity of insects by providing more niches. However, the insects are not restricted to one grass species, but can feed on multiple species. In that case, a niche might be a plant community rather than a grass species, and the grassland expansion might have provided geographically larger niches without increasing their numbers. I expect this would increase population sizes of the moth, which might actually decrease opportunities for allopatric speciation by connecting grasslands spread across Africa. In such a case, the decreased diversification rate of Sesamiina would not be extremely surprising. Indeed, the 'diversification of panicoid grasses leading to diversification of their hosts' might work only in a system where sympatric speciation following host shifts predominate. Can the authors comment on the mechanisms that would be expected to increase moth diversification following the expansion of their hosts?

3) The conclusions of the authors are based on a relatively small group of insects (~200 species). By contrast, the increased diversification of grasses is based on ~10,000 species (when considering the whole family), and analyses of specific groups might have led to different conclusions. Therefore, one can wonder whether the same conclusions would be reached if a wider range of herbivore insects was

considered. Do the authors think that their results are relevant to all C4 grassland-associated insects or do they think that the Sesamiina are a special case in a broad context of herbivore diversity increasing after the expansion of C4 grasslands?

SPECIFIC POINTS:

4) Page 5, line 107: The statement that 'most species [of moth] feed on one or a few distinct plant species' needs to be clarified. In the methods, insect preference is assessed at the family or subfamily level, so that it is not clear whether insect species are indeed restricted to one/few species. If they can feed on larger groups (e.g. a tribe of grasses), then the diversification of grasses might not be very important to determine the number of niches (see my point 2 above). But if they are indeed restricted to one or a few species, references (or data) need to be given to support this statement. I would then deduce that the niches are far from being filled since there are 200 species of moths for 1,100 *Andropogoneae*. I think this needs to be clarified, as this is central to the theses being developed here.

5) Page 9, lines 267-268: I did not understand how this pattern could be identified. The main lineages of C4 grasses originated 20-25 Ma, which is after the Oligocene drop of CO₂. In other words, the whole of their history took place in a world where the CO₂ concentration oscillated between 180 and 300 ppm (way below the 1000-1500 from 32-50 Ma). Therefore, how can CO₂ concentrations explain the diversification of these two C4 lineages?

6) Page 9, lines 273-274: not all C4 grasses benefit from drier conditions. C4 is advantageous in warm, high-light habitats, but most C4 plants need large water supplies during the growing season (many occur in wetlands).

7) Page 10, lines 276-281: The moths are associated with wetlands (and later on, river banks). Did such habitats really decrease in abundance following global cooling? Would global coolings decrease the numbers of rivers and wetlands? Can references be given?

8) Page 10, lines 307-309: See my points 2 and 4 above. If the moths are indeed restricted to one (few) species, then the niches are not saturated (1100 grass species form 200 moths, without mentioning the other groups of C4 grasses and sedges). If they are associated with larger groups, then the number of niches does not change as C4 grasses diversify.

9) Page 11, lines 340-341: echoing my point 2 above, I wonder what mechanism would lead to drier environments limiting the diversification of *Sesamiina*. A reduction of wet habitats would lead to smaller population sizes, but why would this decrease the rate of speciation (small, isolated populations might promote allopatric speciation)?

10) Page 12, lines 373-375: The grass phylogeny is based mostly on plastid markers, while the moth phylogeny is based on equal amounts of mitochondrial and nuclear markers. Organellar markers tend to have more recent coalescent times, which might lead to more recent estimates of divergence, therefore indicating increased diversification. Would the results stay the same if the dating of moth was done using only mitochondrial markers, which are more comparable to those used for grasses?

11) Page 13, line 385: Was the topology fixed in Beast analyses? Else, why would the topology be first inferred with maximum likelihood?

Reviewer #2:

Remarks to the Author:

In this paper, the authors in impressive detail investigate the diversification and possible co-diversification of a group of grass-feeding moths and their associated host grasses. The latter clade, used by most species in the moth clade, is a group of C4 grasses representative of the historical spread of Neogene C4 grasslands. The authors convincingly show that the origins of the two clades in Africa more or less coincide in time, setting the stage for linked adaptive radiations where speciation in the grasses subsequently create new niches for the moths, a classical theory for grass-herbivore coevolution. However, the authors find a different and very interesting pattern where diversification rates in the two lineages over time are not positively but rather negatively correlated. Through the incorporation of environmental proxies in the analyses they show that the opposing patterns are most likely observed because climatic variation has affected the two lineages very differently. The expansion of grasslands indeed seems to have led to an adaptive radiation in the moths, but diversification rates then slowed over time as the niche space filled up in a density-dependent fashion and cooling temperatures simultaneously led to less of the wet niches preferred by the moths being available. In contrast, diversification in the grasses increased over time as the cooling temperatures led to more arid niches (where C4 grasses thrive) being available. Despite more grass niches in principle thus being available for the moths, diversification rates decreased, which is strong evidence for decoupled diversification constrained by the abiotic environment. The fact that similar patterns have recently been reported for e.g. the diversification of horses does not impact negatively on the novelty of the present paper but should rather increase the interest in it by the demonstration of a pattern common to mammals and insects feeding on grasses.

The specimen/species sampling and the ecological data gathered here is very impressive. Although I am not an expert in the analytical methods used, they seem to be state-of-the-art and used appropriately. The reasoning in the paper is well laid out and convincing.

I have only a few minor comments regarding the oscillation hypothesis (OH) which was proposed by myself and Niklas Janz. First, although the "musical chairs" hypothesis was presented as a contrast to the OH (by its authors and here, rows 110-116), we see them as rather similar in that they both predict more species in clades with more "lability" in host use. The real contrast is with hypotheses that focus on adaptive radiations after host shifts, i.e. within the new host clade rather than on a diverse set of hosts. That being said, it is interesting to note that the analysis recovered patterns more consistent with oscillations in host range and re-colonization of ancestral hosts than with host shifts. Second, the citation on row 164 is in error, as we in fact write that host expansions do NOT happen with regular frequency and amplitude, i.e. the term "oscillation" was not well chosen. Third, I am not surprised that the analysis did not pick up a higher speciation rate in clades with more diverse host use. The patterns that we have observed in butterflies and tussock moths are not dramatic enough to be picked up by such methods. Rather, we see a consistent pattern of (often only slightly) more species in the clade using more types of hosts (and often also containing polyphagous species) compared to its more specialized sister clade. It seems like the same consistent pattern is present here. In Fig. S5, compare for instance the two lower-most clades.

Sören Nylin

Reviewer #3:

Remarks to the Author:

The manuscript investigates the role played by grasslands in the diversification of the stemborer group Sesamiina. The authors used this interaction to test a long standing hypothesis about the ecological and evolutionary importance of grasslands in the diversification of herbivore groups.

The manuscript is well-written and the question is clearly of interest. I am however a bit skeptical that the group of insect selected can really test the hypothesis put forward in the manuscript. The clear association between the Sesamiina and the panicoids grasses is interesting, but it is clear that the hygrophylous characteristics of the Sesamiina strongly limit their ability to use the typical grasslands as their main habitat. The negative correlation observed between the diversification of Sesamiina and panicoids can thus not be used to demonstrate the absence of a role played by grasses, because the Sesamiina are simply not adapted to the main grassland habitat. I therefore do not see how the data presented here can test the hypothesis proposed.

One of the key element to link the diversification of Sesamiina and panicoids is the divergence time estimates. You clearly explained the limitations of the panicoids calibrations, but you do not say much about the three secondary calibrations. As shown by several authors in the last few years, it is essential to have calibrations that are congruent between each others. You should therefore test if this is the case using posterior predictive assessments of the calibrations. Further, using secondary calibrations is always difficult because you do not control the parameters that were used in the previous studies. I would strongly suggest to estimate yourself, using a small but representative dataset of noctuids, the divergence time estimates of the nodes that are of interest for you by adding representative Sesamiina sequences. This is alway better that taking publised estimates as granted.

You used family or subfamily levels to estimate the ancestral host-plant associations. This is clearly a limitation, although I understand that it is difficult to be more precise in the assignment of host association. What would be interesting is to clearly know if the number of panicoids species involved in the interactions is representative of the diversity of panicoids or if they only include a small fraction of it. I guess it is the latter, given the results obtained, but a discussion and data on this would be useful. You further mention the Cyperaceae and Typhaceae (line 394), but no results are shown or discussed on these groups. I guess, given their higher link with humid habitats, that the Sesamiina show a positive correlation with their diversification? I would strongly recommend to include them in the results and shift the focus of the manuscript on what caused the diversification of the stem-borer group rather than on the non-effect of grasslands.

Minor comments:

- line 379: how did you set the partitions and how were the models estimated? It is not indicated here except in the Appendix. However, a minimal description should be given here.
- line 393-395: the sentence is not complete.

Reviewers' comments:

Reviewer #1 (Remarks to the Author):

This paper explores the hypothesis that the global expansion of C₄ grasslands during the Miocene allowed the diversification of C₄ grass herbivores. As a study system, it uses the Sesamiina, a group of moths feeding on C₄ plants, and especially panicoid grasses of the subtribe Andropogoneae (the group of grasses that dominates tropical grasslands). Using phylogenies for the host and herbivores, it shows that both groups emerged at the same time around 20 Ma. However, while the diversification rate of Andropogoneae increased through time, that of Sesamiina decreased through time, leading to an apparent inverse correlation. Palaeoclimate models then identify different drivers of diversification in the two groups, with temperature correlating positively with speciation in the insects, but negatively in the Andropogoneae grasses. The authors conclude that different factors dictate the diversification rates of different groups, so that grassland expansion did not universally lead to the diversification of the associated herbivores. This paper is analytically solid, and explores an important issue; the drivers of diversification. It can therefore make a strong contribution to the literature, although there are some conceptual issues that I think require clarification, through additional analyses and/or clarification in the text. I list these below.

We would like to thank the reviewer for his very constructive and thorough review. In our reply we provide detailed clarifications and justifications in response to all his comments, backed up by several supplementary analyses and additional information when necessary.

GENERAL POINTS:

1) *The authors focus on the Andropogoneae grasses, although other groups of grasses and sedges are consumed by the moths. Since the Andropogoneae are the main target of the insects, this can be justified. However, if I understood well, the analyses consider the total diversity of Andropogoneae without consideration for the proportion of these that can serve as hosts for the insects. I could imagine a situation where only a fraction of the Andropogoneae can feed the insect, so that the total diversity of the group is a poor indicator of the niches available for the moths. While these can include species-specific properties, an obvious condition is the geographic distribution. Since the moths are distributed in Africa, the diversity of Andropogoneae in Africa is the only thing that matters, but the diversification analyses consider the overall Andropogoneae distribution. Can the authors verify that the African diversity of Andropogoneae increased at the same pace as the overall diversity of this widely distributed group?*

As highlighted in both the **Results** and **Discussion** parts of our manuscript we are actually focusing on the subfamily **Panicoideae**, not on the tribe **Andropogoneae**. Indeed our ancestral state estimation (ASE) analyses have revealed a high level of phylogenetic niche conservatism on lineages of panicoid grasses as a whole. That said we understand that reviewer#1 was asking for more information on panicoid distribution and diversification. To account for his concern we did the following:

We determined the number of African panicoid species by extracting species distributions from the GRASSWORLD database (<http://grassworld.myspecies.info/en>), using the curated checklist of African grasses available at:

<http://grassworld.myspecies.info/sites/grassworld.myspecies.info/files/Africa.doc>

An estimated total of 1,131 African panicoid species was determined using this checklist in combination with the list of panicoid genera published in the worldwide phylogenetic classification of the Poaceae by Soreng *et al.* (2015) (doi: 10.1111/jse.12150), which also relies on the GRASSWORLD database. Although the African GRASSWORLD species list aims at only including native species it is still a work in progress so we used additional filters to better account for the occurrence of non-native species. First, we removed the 29 species belonging to genera that are known to be non-native to the Afrotropical region (Soreng *et al.*, 2015). Second, as a way to detect potentially non-native species, we removed the 67 most widespread species (hereby defined as distributed in more than 20 African countries). Finally we removed seven species that were exclusively found in North Africa (which is part of the Palearctic biogeographic realm). As a result we ended up with an estimate of **1,034** native panicoid species for the Afrotropical region, to be compared with a total species richness of **3,560** species for the whole subfamily (Soreng *et al.*, 2015). Because several non-native species are still potentially included in our list (native distributions are sometime ambiguous) we think that the number of native panicoid species is actually comprised between 900 (conservative low boundary) and 1,034 species.

We then used these two diversity estimates to perform supplementary analyses using the Binary State Speciation Extinction (BiSSE) model (Maddison *et al.*, 2007; doi: 10.1080/10635150701607033). This model was used to estimate the respective diversification rates of Afrotropical and non-Afrotropical panicoid species in order to assess whether “*African diversity of Andropogoneae increased at the same pace as the overall diversity of this widely distributed group*”. To do so we carried out eight BiSSE models, from the null model (no variation of rates) to the most complex model in which all parameters of speciation, extinction and transition are estimated. We considered a conservative Afrotropical species number sampled in the phylogeny (*i.e.*, 155 species) by excluding all species not exclusively distributed in the Afrotropics. The most complex model (*all.free*) was defined as the best-fit model by the AICc comparisons (see the table below). Whatever the total Afrotropical diversity estimates are (900 vs. 1,034), for both groups we found **very similar net diversification rates** ($r = \lambda - \mu$) comprised between 0.24 and 0.28 events/Myr/lineage (0.24-0.25 for the Afrotropical native Panicoideae, vs. 0.28 for the remaining panicoid species). These analyses (now presented in **Appendix S1**) thus strongly support the hypothesis that the Afrotropical diversity of Panicoideae increased at the same pace as the overall diversity of other members of the subfamily.

BiSSE analyses with an estimated diversity of 900 species for the Afrotropical panicoid species							
Model	NP	logL	AICc	lambda0	lambda1	mu0	mu1
Null model	3	-2355.263	4716.556	NA	0.7692	NA	0.5099
lambda.free	4	-2346.041	4700.131	0.7454	0.5909	NA	0.414
mu.free	4	-2345.351	4698.751	NA	0.7192	0.3891	0.5523
q.free	4	-2345.377	4698.804	NA	0.7734	NA	0.5154
lambda.mu.free	5	-2345.296	4700.668	0.7225	0.6876	0.3907	0.5193
lambda.q.free	5	-2341.706	4693.487	0.7546	0.6576	NA	0.4516
mu.q.free	5	-2342.989	4696.053	NA	0.7463	0.4482	0.5453
all.free	6	-2340.138	4692.382	0.8147	0.5028	0.5312	0.2454

BiSSE analyses with an estimated diversity of 1,034 species for the Afrotropical panicoid species							
Model	NP	logL	AICc	lambda0	lambda1	mu0	mu1
Null model	3	-2350.249	4706.527	NA	0.7544	NA	0.4924
lambda.free	4	-2343.462	4694.973	0.7348	0.6017	NA	0.409
mu.free	4	-2342.66	4693.37	NA	0.7097	0.3842	0.526
q.free	4	-2343.125	4694.3	NA	0.7593	NA	0.4991
lambda.mu.free	5	-2342.658	4695.391	0.7135	0.7102	0.3888	0.527
lambda.q.free	5	-2340.63	4691.336	0.7445	0.6627	NA	0.4453
mu.q.free	5	-2341.275	4692.625	NA	0.7317	0.4334	0.521
all.free	6	-2340.196	4692.498	0.7754	0.5745	0.4872	0.3309

2) The authors hypothesize that the diversity of grasses would increase the diversity of insects by providing more niches. However, the insects are not restricted to one grass species, but can feed on multiple species. In that case, a niche might be a plant community rather than a grass species, and the grassland expansion might have provided geographically larger niches without increasing their numbers. I expect this would increase population sizes of the moth, which might actually decrease opportunities for allopatric speciation by connecting grasslands spread across Africa. In such a case, the decreased diversification rate of *Sesamiina* would not be extremely surprising. Indeed, the 'diversification of panicoid grasses leading to diversification of their hosts' might work only in a system where sympatric speciation following host shifts predominate. Can the authors comment on the mechanisms that would be expected to increase moth diversification following the expansion of their hosts?

We disagree here because *Sesamiina* stemborers actually exhibit a very high level of host-specialization: most species are only feeding on one or a few grass species (e.g., see Le Ru et al., 2006; doi: [10.1080/00379271.2006.10697467](https://doi.org/10.1080/00379271.2006.10697467)). To illustrate this point we have updated and expanded the **supplementary Table S2** in order to list all extant host-plant records; in this revised appendix we also present graphs that illustrate the level of host-specialization of *Sesamiina* stemborers at the (plant) genus and species level. These graphs (see below) clearly illustrate our point as most *Sesamiina* species are associated with three or less plant genera (94%) or species (88%).

This high level of host-specialization - far from a 'plant community' level - does not come out as a complete surprise because it is a pattern that is consistently observed in other groups of specialized endophagous herbivores (Bernays & Chapman, 1994; [doi: 10.1007/b102508](https://doi.org/10.1007/b102508); Gaston *et al.*, 1992; [doi: 10.1111/j.1365-2311.1992.tb01044.x](https://doi.org/10.1111/j.1365-2311.1992.tb01044.x); Kergoat *et al.*, 2017; [doi: 10.1016/bs.abr.2016.09.005](https://doi.org/10.1016/bs.abr.2016.09.005)) such as curculionine weevils (Marvaldi *et al.*, 2002; [doi: 10.1080/10635150290102465](https://doi.org/10.1080/10635150290102465)) or bruchine seed-beetles (Kergoat *et al.*, 2008; ISBN: 9789004152045).

In addition - and to respond to the reviewer's comment regarding the potential role of allopatric/sympatric speciation - we have added in the same appendix additional information on the ecology of Sesamiina stemborers, especially in reference to their distribution. We think that we cannot exclude the role of allopatric speciation (in addition to sympatric speciation) because the vast majority of Sesamiina species have a restricted geographic range (*e.g.*, Le Ru *et al.*, 2015; [doi: 10.11646/zootaxa.3925.1.4](https://doi.org/10.11646/zootaxa.3925.1.4); Le Ru *et al.*, 2017a; [doi: 10.1080/00379271.2017.1320586](https://doi.org/10.1080/00379271.2017.1320586); Le Ru *et al.*, 2017b; [doi: 10.1080/00379271.2017.1332959](https://doi.org/10.1080/00379271.2017.1332959)); for instance numerous species are associated with a specific bioregion or specific mountain ranges (*e.g.*, Uzungwa mountains, Drakensberg range, Kipengere range). Species that supposedly had large distribution areas also generally correspond to species complexes.

3) *The conclusions of the authors are based on a relatively small group of insects (~200 species). By contrast, the increased diversification of grasses is based on ~10,000 species (when considering the whole family), and analyses of specific groups might have led to different conclusions. Therefore, one can wonder whether the same conclusions would be reached if a wider range of herbivore insects was considered. Do the authors think that their results are relevant to all C4 grassland-associated insects or do they think that the Sesamiina are a special case in a broad context of herbivore diversity increasing after the expansion of C4 grasslands?*

First, and as underlined by our species count for Afrotropical panicoid species, the difference in term of species richness between the two groups is not that drastic; it seems more relevant to contrast the number of stemborers species (~200 species) with the diversity of native Afrotropical panicoid species (estimated here at ~900-1,034 species).

Second, we tend to think that Sesamiina stemborers are likely representative of other insect lineages which exhibit a similar high level of specialization on C₄ grasses. For example we can make a parallel with other C₄-specialized lineages such as anisopliine beetles (Mico *et al.*, 2009; [doi: 10.1111/j.1365-2699.2008.02010.x](https://doi.org/10.1111/j.1365-2699.2008.02010.x)) or derived lineages of Delphacini planthoppers (Urban *et al.*, 2010; [doi: 10.1111/j.1365-3113.2010.00539.x](https://doi.org/10.1111/j.1365-3113.2010.00539.x)) and Satyrini butterflies (Aduse-Poku *et al.*, 2015; [doi: 10.1186/s12862-015-0449-3](https://doi.org/10.1186/s12862-015-0449-3)), which have also diversified in the last 25 Myrs and exhibit comparable levels of species richness (between 100 to 300 species).

SPECIFIC POINTS:

4) *Page 5, line 107: The statement that 'most species [of moth] feed on one or a few distinct plant species' needs to be clarified. In the methods, insect preference is assessed at the family or subfamily level, so that it is not clear whether insect species are indeed restricted to one/few species. If they can feed on larger groups (e.g. a tribe of grasses), then the diversification of grasses might not be very important to determine the number of niches (see my point 2 above). But if they are indeed restricted to one or a few species, references (or data) need to be given to support this statement. I would then deduce that the niches are far from being filled since there are 200 species of moths for 1,100 Andropogoneae. I think this needs to be clarified, as this is central to the theses being developed here.*

As underlined previously - to answer this comment and the comment above - we have expanded the **supplementary Table S2** and also provide two graphs that show precisely the level of host-specialization in Sesamiina stemborers (please see our detailed answer to point #2).

5) Page 9, lines 267-268: *I did not understand how this pattern could be identified. The main lineages of C₄ grasses originated 20-25 Ma, which is after the Oligocene drop of CO₂. In other words, the whole of their history took place in a world where the CO₂ concentration oscillated between 180 and 300 ppm (way below the 1000-1500 from 32-50 Ma). Therefore, how can CO₂ concentrations explain the diversification of these two C₄ lineages?*

First we would like to underline that at the origin of both groups, CO₂ concentrations were around 600 ppm (please see **Fig. 2F** or the reference study from Beerling & Royer, 2011; doi: [10.1038/ngeo1186](https://doi.org/10.1038/ngeo1186)), and not 300 ppm as suggested by the reviewer. Hence the impact of the drop of CO₂ is potentially way higher.

Second CO₂ was not recovered as positively correlated with the diversification of both groups (as suggested by the reviewer's comment); it was only positively correlated with the diversification of C₄ grass lineages, which make sense because C₄ plants are better adapted to lower atmospheric CO₂ concentrations thanks to a series of anatomical and biochemical modifications that concentrate CO₂ around the carboxylating enzyme Rubisco, thereby increasing photosynthetic efficiency in conditions promoting high rates of photorespiration, such as arid conditions (Sage, 2004; doi: [10.1111/j.1469-8137.2004.00974.x](https://doi.org/10.1111/j.1469-8137.2004.00974.x)).

6) Page 9, lines 273-274: *not all C₄ grasses benefit from drier conditions. C₄ is advantageous in warm, high-light habitats, but most C₄ plants need large water supplies during the growing season (many occur in wetlands).*

We agree on the fact that not all C₄ grass species benefit from drier conditions. However our point here is to say that most C₄ grass species have a clear advantage over C₃ grass species when facing drier conditions, especially in relation with drought resistance (see e.g., Taylor *et al.*, 2014; doi: [10.1111/gcb.12498](https://doi.org/10.1111/gcb.12498)). To account for this remark we modified the corresponding sentence accordingly (see below) and added the corresponding reference (Taylor *et al.*, 2014) in relation to advantage conferred by the resistance to drought.

'From a biological point of view C₄ grasses are less drought sensitive than C₃ grasses⁶⁰, and generally benefit from drier conditions associated with high-light habitats!.'

7) Page 10, lines 276-281: *The moths are associated with wetlands (and later on, river banks). Did such habitats really decrease in abundance following global cooling? Would global cooling decrease the numbers of rivers and wetlands? Can references be given?*

There is a documented decrease of wetland habitats - shift from moist woodlands/wetlands with predominantly summer rainfall to a range of more arid to semi-arid environments with increasingly seasonal rainfall regimes - since the Oligocene, especially after the mid-Miocene climatic optimum (e.g., deMenocal, 2004; doi: [10.1016/S0012-821X\(04\)00003-2](https://doi.org/10.1016/S0012-821X(04)00003-2); Bob, 2006; doi: [10.1016/j.jaridenv.2006.01.010](https://doi.org/10.1016/j.jaridenv.2006.01.010); Sepulchre *et al.*, 2006; doi: [10.1126/science.1129158](https://doi.org/10.1126/science.1129158); Udeze & Oboh-Ikuenobe, 2005; doi: [10.1016/j.palaeo.2004.12.026](https://doi.org/10.1016/j.palaeo.2004.12.026)). To account for the reviewer's remark we modified the corresponding sentence accordingly (see below) and provide additional references in the revised version of the main text of the manuscript.

'Interestingly, changes in temperature over time (Fig. 2F) are also associated with changes in climatic conditions: during the Cenozoic, warmer periods have tended to be associated with wetter conditions, and colder periods are associated with drier conditions,

with a shift from moist woodlands/wetlands and predominantly summer rainfall to a range of more arid to semi-arid open habitats and increasingly seasonal rainfall regimes^{57,58,59}.

8) Page 10, lines 307-309: *Seem my points 2 and 4 above. If the moths are indeed restricted to one (few) species, then the niches are not saturated (1100 grass species form 200 moths, without mentioning the other groups of C₄ grasses and sedges). If they are associated with larger groups, then the number of niches does not change as C₄ grasses diversify.*

The results of our diversity-dependent diversification analyses suggest the existence of ecological limits on moth diversification, which implies the saturation of potential niches (*i.e.*, the C₄ grass species). Although the total number of potential host-plants is elevated, not all grass species are equally suitable: (i) some grass species have silicon concentrations that are too high for the stemborer larvae to cope with (Calatayud *et al.*, 2016; doi: [10.1016/j.agee.2016.03.040](https://doi.org/10.1016/j.agee.2016.03.040)), (ii) some grass species have stems that are too small to accommodate all larval stages, and (iii) some grass species are only found in dry environments, which influence plant form and phytolith formation (please see our response to point #9 just below). All of this translates into a reduction of the total number of suitable niches (*i.e.*, realized vs. fundamental niches) for the stemborer moths. While considering the high level of specialization of stemborers it is thus possible to have a saturation of (realized)niches despite the fact that C₄ grass species are five time more diverse than the stemborers. To precise this point (and also point #9) we added the following paragraph in the revised version of the main text of manuscript:

'The marked preference of Sesamiina for wet habitats may have limited the dimensions of their available niche space in an increasingly dry environment. Dry environments also directly influence plant form and phytolith formation in C₄ grasses. As underlined by a recent large scale study on West African grasses⁶³, drier environments are characterized by smaller, more therophytic grass species, which are too small to accommodate all stemborer larval stages. Sesamiina larvae are also sensitive to the high levels of silicon content⁶⁴. One of the factors promoting phytolith formation (accumulation of silicon) in grasses is evapotranspiration⁶⁵, which is more important in drier environments⁶⁶. All of these observations are consistent with a reduction of the availability of suitable niches in dry environments and may reflect or even explain the preference of Sesamiina for wet habitats.'

9) Page 11, lines 340-341: *echoing my point 2 above, I wonder what mechanism would lead to drier environments limiting the diversification of Sesamiina. A reduction of wet habitats would lead to smaller population sizes, but why would this decrease the rate of speciation (small, isolated populations might promote allopatric speciation)?*

For the diversification of Sesamiina we saw two limiting factors associated with drier environments, which directly relates to our answer of point #8.

Silicon content. Sesamiina stemborers are sensitive to the level of silicon content (Calatayud *et al.*, 2016). One of the factors promoting phytolith formation (accumulation of silicon) in grasses is evapotranspiration (Takahashi *et al.*, 1990), the later of which is more important in drier environments (Sage, 2004; Issaharou-Matchi *et al.*, 2016; doi: [10.1016/j.flora.2016.02.008](https://doi.org/10.1016/j.flora.2016.02.008)).

Plant form. As underlined by a recent large scale study on West African grasses (Schmidt *et al.*, 2016; doi: [10.1111/j.1365-2028.2011.01283.x](https://doi.org/10.1111/j.1365-2028.2011.01283.x)), drier environments are characterized by smaller, mainly therophytic grass species while more humid environments are characterized by larger, often hemicyrptophytic, grass species. The later thus constitute more appropriate hosts for the stemborers, both in term of size and availability.

Because of these two limiting factors we think that grasses growing in dried environments constitute less suitable hosts for the stemborers. This echoes our reply (point #8) on the difference between the number of potential niches for the insects (fundamental vs. realized niches).

10) Page 12, lines 373-375: *The grass phylogeny is based mostly on plastid markers, while the moth phylogeny is based on equal amounts of mitochondrial and nuclear markers. Organellar markers tend to have more recent coalescent times, which might lead to more recent estimates of divergence, therefore indicating increased diversification. Would the results stay the same if the dating of moth was done using only mitochondrial markers, which are more comparable to those used for grasses?*

Both molecular datasets (grasses and insects) are actually quite comparable in terms of sampling of organellar and nuclear genes. Our dataset consists of four mitochondrial genes and two nuclear markers. The grass dataset has a lot of heterogeneity (major grass groups are represented by distinct set of genes) with a lot of missing data. It nonetheless includes six nuclear markers (see Table S2 in Spriggs *et al.*, 2014; [doi: 10.1371/journal.pone.0097722](https://doi.org/10.1371/journal.pone.0097722)); though it does encompass more chloroplastic fragments (up to 29) most of them are overlapping so the actual ratio is actually close to ours.

Moreover, organellar markers do NOT have more recent coalescent times; actually this is the contrary. There are multiple studies demonstrating that even with identical calibration regimes and molecular clock methods, mitochondrial organellar-only based molecular age estimates are systematically older than those estimated from nuclear-only data (e.g., Dornburg *et al.*, 2014; [doi: 10.1186/s12862-014-0169-0](https://doi.org/10.1186/s12862-014-0169-0); Marshall *et al.*, 2016; [doi: sysbio/article/65/1/16/2461540](https://doi.org/10.1186/s12862-016-2461-5)). Therefore if the dating of moths was done using only mitochondrial markers we think that we would have had an even more marked (but also artefactual) pattern of decreasing diversification with even less speciation events in recent times.

11) Page 13, line 385: *Was the topology fixed in Beast analyses? Else, why would the topology be first inferred with maximum likelihood?*

We used a fixed topology, which is a standard procedure (and often the only way) to achieve a good convergence of MCMC runs (with ESS values > 200). We relied on the maximum likelihood tree because it was the one with the highest level of support; again this is a quite standard procedure in dating analyses. All this information was already presented in the original version of the supplementary material.

Reviewer #2 (Remarks to the Author):

In this paper, the authors in impressive detail investigate the diversification and possible co-diversification of a group of grass-feeding moths and their associated host grasses. The latter clade, used by most species in the moth clade, is a group of C₄ grasses representative of the historical spread of Neogene C₄ grasslands.

The authors convincingly show that the origins of the two clades in Africa more or less coincide in time, setting the stage for linked adaptive radiations where speciation in the grasses subsequently create new niches for the moths, a classical theory for grass-herbivore coevolution. However, the authors find a different and very interesting pattern where diversification rates in the two lineages over time are not positively but rather negatively correlated.

Through the incorporation of environmental proxies in the analyses they show that the opposing patterns are most likely observed because climatic variation has affected the two lineages very differently. The expansion of grasslands indeed seems to have led to an adaptive radiation in the moths, but diversification rates then slowed over time as the niche space filled up in a density-dependent fashion and cooling temperatures simultaneously led to less of the wet niches preferred by the moths being available. In contrast, diversification in the grasses increased over time as the cooling temperatures led to more arid niches (where C₄ grasses thrive) being available. Despite more grass niches in principle thus being available for the moths, diversification rates decreased, which is strong evidence for decoupled diversification constrained by the abiotic environment.

The fact that similar patterns have recently been reported for e.g. the diversification of horses does not impact negatively on the novelty of the present paper but should rather increase the interest in it by the demonstration of a pattern common to mammals and insects feeding on grasses.

We want to thank the Sören Nylin for his detailed appreciation of our work and for comments that helped improving key-points on the hypotheses related to insect-plant interactions. We are also very grateful for the very positive input on the study overall.

The specimen/species sampling and the ecological data gathered here is very impressive.

We really appreciate that Sören Nylin underlined this point. We (and especially Bruno who literally spent years on the field) have indeed made a special effort to include the largest species-level sampling to estimate accurately all aspects of the evolutionary history of the group. Up to our knowledge such dataset is also unique for the Afrotropical region ... and likely will remain unique for a long time due to safety risks associated with field collecting in several of the prospected countries.

Although I am not an expert in the analytical methods used, they seem to be state-of-the-art and used appropriately. The reasoning in the paper is well layed out and convincing.

We also think that it is one of the strengths of this study.

I have only a few minor comments regarding the oscillation hypothesis (OH) which was proposed by myself and Niklas Janz. First, although the "musical chairs" hypothesis was presented as a contrast to the OH (by its authors and here, rows 110-116), we see them as rather similar in that they both predict more species in clades with more "lability" in host use. The real contrast is with hypotheses that focus on adaptive radiations after host shifts, i.e. within the new host clade rather than on a diverse set of hosts. That being said, it is interesting to note that the analysis recovered patterns more consistent with oscillations in host range and re-colonization of ancestral hosts than with host shifts.

We want to thank Sören Nylin for these very insightful comments. To account for them we made two changes in the main text. First we added the following sentence in the **Introduction** section:

‘Both hypotheses are similar in that they predict more species in clades with more ‘lability’ in host use (Nylin pers. com.).’

Second we removed the following assertion from the **Results** section:

‘Contrary to the prediction of the OH ...’

Second, the citation on row 164 is in error, as we in fact write that host expansions do NOT happen with regular frequency and amplitude, i.e. the term "oscillation" was not well chosen.

This has been corrected.

Third, I am not surprised that the analysis did not pick up a higher speciation rate in clades with more diverse host use. The patterns that we have observed in butterflies and tussock moths are not dramatic enough to be picked up by such methods. Rather, we see a consistent pattern of (often only slightly) more species in the clade using more types of hosts (and often also containing polyphagous species) compared to its more specialized sister clade. It seems like the same consistent pattern is present here. In Fig. S5, compare for instance the two lower-most clades.

This is a very interesting remark which echoes a potential future project of ours, which implies carrying a detailed study on the remaining members of the tribe Apameini (from subtribe Apameina), which are generally more polyphagous and more diverse (but also older and with an holarctic distribution).

Reviewer #3 (Remarks to the Author):

The manuscript investigates the role played by grasslands in the diversification of the stemborer group Sesamiina. The authors used this interaction to test a long standing hypothesis about the ecological and evolutionary importance of grasslands in the diversification of herbivore groups.

The manuscript is well-written and the question is clearly of interest. I am however a bit skeptical that the group of insect selected can really test the hypothesis put forward in the manuscript. The clear association between the Sesamiina and the panicoids grasses is interesting, but it is clear that the hygrophilous characteristics of the Sesamiina strongly limit their ability to use the typical grasslands as their main habitat. The negative correlation observed between the diversification of Sesamiina and panicoids can thus not be used to demonstrate the absence of a role played by grasses, because the Sesamiina are simply not adapted to the main grassland habitat. I therefore do not see how the data presented here can test the hypothesis proposed.

Sesamiina are unquestionably adapted to the main grassland habitat as attested by the fact that the great majority of stemborer species were collected with light-traps set in open habitats (we now precise this point in different parts of the supplementary material).

The nuance here relates to the fact that numerous hygrophilous species are found in contact zones between open grasslands and wetlands/marshes/wet forests. It is especially the case in low altitude areas. It is less the case at mid-higher elevation where stemborer species are only found in open grassland habitats, such as the Veld in South Africa or mountain grasslands in the East African mountain ranges.

Please note that all this information is now presented in the revised **Table S2**.

One of the key element to link the diversification of Sesamiina and panicoids is the divergence time estimates. You clearly explained the limitations of the panicoids calibrations, but you do not say much about the three secondary calibrations. As shown by several authors in the last few years, it is essential to have calibrations that are congruent between each others. You should therefore test if this is the case using posterior predictive assessments of the calibrations.

The reviewer underlines that it is - *essential to have calibrations that are congruent between each others* -. We are in complete agreement with that; however that is something that usually applies to primary (fossil) calibrations, where several specific procedures are used (for instance fossil cross-validations *sensu* Near & Sanderson, 2004; doi: 10.1098/rstb.2004.1523 / Near *et al.*, 2005; doi: 10.1086/427734) to assess whether distinct fossil calibrations are congruent (please note that the first author is familiar with these procedures; see *e.g.*, Aghová *et al.*, 2017; doi: /10.1101/180398). Because we relied here on **secondary calibrations from the same study**, these constraints are also **intrinsically congruent**; so we really do not see the rationale for using posterior predictive assessments of the calibrations here. In addition it is worth underlining that posterior predictive assessments are generally used in dating analyses to assess clock model adequacy (Duchêne *et al.*, 2015; doi: 10.1093/molbev/msv154; see also the related BEAST tutorial here: https://taming-the-beast.org/tutorials/adequacy_tutorial/).

That said, and to account for the reviewer's comment, we carried out additional BEAST analyses to assess the level of congruence of the three secondary calibrations. To do so we tested the impact of removing a specific constraint on the age estimation of the corresponding node. Files generated by BEAST were processed with TRACER, which allowed us to generate detailed outputs (here we only present one of them).

For the secondary calibration set on the stem node of the genus *Condica*, the BEAST analysis not relying on this constraint recovers a highly congruent age estimate of 31.41 Ma (95% HPD: 27.05-35.16) to be compared with median age of 31.50 Ma (95% HPD: 27.48-34.78) for the analysis with all three secondary calibrations.

For the secondary calibration set on the stem node of the genus *Cryphia*, the BEAST analysis not relying on this constraint recovers a congruent age estimate of 30.67 Ma (95% HPD: 25.57-36.81) to be compared with median age of 28.86 Ma (95% HPD: 25.63-31.18) for the analysis with all three secondary calibrations. Finally, for the secondary calibration set on the root, the BEAST analysis not relying on this constraint recovers a congruent age estimate of 31.27 Ma (95% HPD: 27.29-36.66) to be compared with median age of 33.58 Ma (95% HPD: 31.63-38.20) for the analysis with all three secondary calibrations.

These additional analyses indicate that age estimates are congruent and do not reveal conflicting secondary calibrations.

Further, using secondary calibrations is always difficult because you do not control the parameters that were used in the previous studies. I would strongly suggest to estimate yourself, using a small but representative dataset of noctuids, the divergence time estimates of the nodes that are of interest for you by adding representative Sesamiina sequences. This is always better than taking published estimates as granted.

Regarding the second suggestion made by the reviewer - *estimate yourself, using a small but representative dataset of noctuids, the divergence time estimates of the nodes that are of interest for you by adding representative Sesamiina sequences* - it basically corresponds to the BEAST analyses we did in 2012 (Toussaint *et al.*, 2012; doi: 10.1371/journal.pone.0041377). In this study we relied on a small but representative noctuid dataset (94 species including 15 Sesamiina) and on a primary fossil calibration of an undetermined (Sohn *et al.*, 2012; <http://www.mapress.com/j/zt/article/view/13332>) and somewhat disputed erebid fossil. Resulting age estimates are very similar (only slightly older), as they yielded median ages of ~23.9 Ma and ~24.0 Ma with a BD and a Yule model, respectively (to be compared with our age estimates of ~20.72 Ma and ~20.63 Ma for the Sesamiina).

Because the status of this erebid fossil is still discussed, for this study we chose to rely on an independent source of calibration constraints. Hence our decision of using secondary calibrations taken from a comprehensive dating study focusing on all lepidopteran groups (350 taxa sampled in Wahlberg *et al.*, 2012; doi: 10.1371/journal.pone.0080875), which relies on well-documented lepidopteran fossils from groups unrelated to the noctuids. Ultimately the fact that the age estimates of our new study are congruent with those of Toussaint *et al.* (2012) provides more support for our work.

You used family or subfamily levels to estimate the ancestral host-plant associations. This is clearly a limitation, although I understand that it is difficult to be more precise in the assignment of host association. What would be interesting is to clearly know if the number of panicoid species involved in the interactions is representative of the diversity of panicoids or if they only include a small fraction of it. I guess it is the latter, given the results obtained, but a discussion and data on this would be useful. You further mention the Cyperaceae and Typhaceae (line 394), but no results are shown or discussed on these groups. I guess, given their higher link with humid habitats, that the Sesamiina show a positive correlation with their diversification? I would strongly recommend to include them in the results and shift the focus of the manuscript on what caused the diversification of the stem-borer group rather than on the non-effect of grasslands.

This is an interesting remark. When studying the evolution of insect-plant associations there is undoubtedly a trade-off between using species or subfamilial/familial levels for the plants. As we pointed out most studies focus on higher taxonomic levels, in order to implement proper statistical treatments (in our case ASE analyses that cannot deal with too many character states) and to limit biases linked to lacks of knowledge on the host-ranges of insect species.

Here we have the advantage of working on internal feeders (which guarantees a correct assignment of the host-plant). Although our assessment of the host-range of all species (now provided with all details in the revised **Table S2**) is likely incomplete (especially for the species associated with undetermined panicoid grasses), we think that we do capture the major evolutionary trends of the group, especially regarding the high level of specialization on C₄ panicoid grasses. Through the years we have consistently added new host-records, and it is always the same story: except for the few (relatively) polyphagous species (such as *Sesamia nonagrioides*), we are always adding new C₄ panicoid grass species to extant host-ranges and the level of host-specialization still remains very high. So the overall pattern remains the same.

Regarding the point on Typhaceae and Cyperaceae, for Typhaceae, only two *Typha* species have been recovered as hosts, so nothing can be really done here in term of diversification analyses (also there are only 51 species in the family, 12 of which belong to *Typha*). For Cyperaceae, only a small sesamiine clade of less than 10 species (genus *Sciomesa*) have a strong association with sedges (as underlined by the results of the ASE). Though the sedge family is very diverse (5,475 species of which about 700 belong to the genus *Cyperus*) very few are used as a host-plants by Sesamiina stemborers so there is clearly no positive correlation with the diversification of both groups.

Minor comments:

- line 379: *how did you set the partitions and how were the models estimated? It is not indicated here except in the Appendix. However, a minimal description should be given here.*

We added this information to the main text of the revised manuscript.

- line 393-395: *the sentence is not complete.*

The corresponding sentence was completed.

In summary, we have taken into account all the comments and corrections brought up by you and the reviewers, including additional the analyses, revising and enhancing the manuscript. We hope you find our revision appropriate and we are looking forward to receiving your response.

Yours sincerely (on behalf of all co-authors)

Gael J. Kergoat

Reviewers' Comments:

Reviewer #1:

Remarks to the Author:

The authors have reworked their manuscript adequately. There are still some imperfections, but I think these are inherent to such analyses of large, complex datasets, and do not in any case affect the validity of the results. I would still like to take the opportunity to reply to some of the authors comments.

1) Their BiSSE analyses do show that Afrotropical and non-Afrotropical have the same net diversification rate 'on average'. That said, it does not directly say whether the changes in diversification rates through time identified by the other analyses happened within Africa or not. Again, I do not think that it affects in any way the overall conclusion of the study, but one could imagine based on the analyses presented in this manuscript that the 'increasing rates of speciation over time' for grasses happened outside of Africa. It might very well have concerned both Afrotropical and non-Afrotropical Panicoideae or just Afrotropical Panicoideae, this is just not known. But I do agree that based on the results presented here as well as the external literature, a decrease of Panicoideae diversity within Africa is very unlikely, so that the conclusions that the hosts and insects diversified in different directions stands.

2) Regarding the host-specificity of Sesamiina, I do not find the new analyses extremely compelling. Am I correct that host preference was established by offering 220 species of grasses and sedges to larvae? If so, the full range of host species able to sustain each insect has not been established (only a few portion of the Afrotropical species were evaluated), especially because a number of the hosts were not identified at the species level according to Table S2. I also note from the referenced study (Le Ru et al. 2006) that up to 11 species grew on the same grass species, and out of 36 insects analysed by Le Ru et al., only 21 (the 24 they report was obtained by considering hosts from the same genus as the same host) were found on only one host species. Because the sampling might have missed alternative host plants, it means that at least 40% of the species in this study used to support a 'high level of host specialization' can have multiple host plants. Overall, I do therefore not find strong evidence for a tight association between insects and hosts. I do moreover think that the ancestral state analysis presented in Fig. S5 is not really informative with this respect as it is based on large groups of hosts, which include hundreds to thousands of species. Again, I do not think that this affects the conclusion that the insects and their hosts diversified at different times.

Overall, after reading the new version of the manuscript and the responses of the authors (and the Le Ru et al. paper suggested by the authors), I think that the lack of association between the diversification of the insects and the grasses is explained by the fact that grass species do not represent distinct ecological niches. First, multiple insect species can prosper on the same grass species. Second, a good fraction of the insects can prosper on multiple grass species (and this fraction is probably underestimated, since not all hosts have been tested). The text added by the authors (lines 398-416) similarly claims that many of the grasses do not represent potential niches for the insects because they are too small for the larvae or because of their high silica content. This therefore explains the disconnect between the diversification patterns of the two groups.

Line 373: 'in this case, C4 grass lineages' → presumably, a grass lineage is not the niche, is it?

Reviewer #2:

Remarks to the Author:

All my comments on the manuscript has now been adressed, and I congratulate the authors on an impressive body of work and a very interesting contribution to research on diversification!

Sören Nylin

Reviewer #3:

Remarks to the Author:

The authors propose a revised version of their manuscript that account for the different comments raised by the reviewers. In particular, they added several new analyses to complete and backup several of their results and modify parts of the manuscript to clarify the text.

I was happy to see the results of the BiSSE analyses showing that diversification rates of the African panicoids are similar to the whole panicoid clade. This is a comment that I did not raise, but it is an important point to address and the analyses done show nicely this point.

I also thank the authors for the new divergence time analyses, which shows that the calibration points used are congruent and robust. It does not, however, alleviate the fact that secondary calibrations coming from other studies might not be appropriate if any biased affected these studies, and that it is better control such calibrations by analysing ourselves the available data. This part of the divergence time analyses should however not affect drastically the main results of the manuscript and I am happy with the controls provided here.

The last point that I have relates to my first comment, which is somewhat also linked with some comments raised by reviewer #1 (e.g. point 8. in minor comments). It is indicated (line 397-398 of the revised manuscript) that the "marked preference [...] for wet habitats may have limited" available niche space for Sesamiina. Similarly, you argue in response to review #1 that Sesamiina have ecological limits on their diversification with potential saturation of the available niche. Recalling your main results, you thus have a large diversification in panicoids, which is not associated with a joint diversification of Sesamiina presumably because of their niche limitation and their preference for wet habitat. It is thus not a counterexample for the role of grasslands in herbivore diversification because the Sesamiina are not adapted to the main conditions found in diversified grasslands, that is dry habitats. To show this, you would need an herbivore group with an ecological niche similar to grasslands that did not follow the same diversification dynamic than grasslands. In contrast, the study nicely show the factors associated with the evolution of this interesting group of insects, but the premise of the manuscript to place their study in the context of the role played by grassland in their evolution is in my opinion flawed. The revised manuscript is not addressing this point and the only way to do it is to change the structure of the introduction and discussion.

Additional comments:

Editor comments (Remarks to the Author):

1) *Diversification patterns in Afrotropical and non-Afrotropical grasses:*

1a- Could you address this issue by removing from the analyses any branches that were likely outside of Africa based on current distributions of extant species and ancestral range reconstruction?

As underlined by the results of the BioGeoBEARS analyses (Figure S6, see below), panicoid grasses have a dynamic biogeographic history with multiple dispersal events from within and outside the Afrotropical region, as well as instances of back-dispersal. As a result there is no such thing as biogeographically distinct exclusively Afrotropical clades that could be analysed separately.

More importantly, it would be fundamentally flawed to conduct diversification analyses relying on the removal of any branches that are likely outside of the Afrotropics. We are not aware of any of examples of such a pruning procedure's being used for diversification analyses. Pruning specific branches will artificially bias the inferred diversification regimes of corresponding clades, and in turn bias the results of potential diversification analysis. It has been shown that undersampling species diversity in dated trees lead to such bias in estimating diversification rates (Heath *et al.* 2008 – Syst. Biol.; Cusimano & Renner 2010 – Syst. Biol.; Brock *et al.* 2011 – Syst. Biol.).

However, an alternative solution for estimating rates of diversification based on trees that imply multiple (homoplasious) acquisitions of a given trait is the implementation of BiSSE analyses specifically tailored to assess such trait- or area-specific differences in diversification regimes (e.g. Goldberg *et al.* 2010 – Science; Rabosky & Glor 2010 – PNAS; Hugall &

Stuart-Fox 2012 – Nature). Having said that - and to alleviate your concerns - in this revised version we provide results of additional BiSSE analyses that provide more support for our results regarding the lack of significant underlying differences in diversification dynamics between Afrotropical and non-Afrotropical lineages (please see our more detailed response below). In reference to this point we also discuss the results of BioGeoBEARS in relation to the results of BAMM diversification analyses; this comparison allows us to highlight that differences in diversification regimes are not associated with biogeographic events (please also see our response to point 1b-).

1b- Rather than presenting diversification rates for the whole clade (and justifying with BiSSE results showing no differences in diversification rate by region), please consider whether it would be more parsimonious to compare diversification rates for only the Afrotropical portion of the clade with the moth diversification rates (e.g. in Figure 2).

To provide more support to our claim regarding the fact that the diversification dynamics of both Afrotropical and non-Afrotropical lineages are similar, we present three new lines of evidence.

1. The first echoes a remark made by reviewer #1 to the effect that our BiSSE analyses only show that Afrotropical and non-Afrotropical have the same net diversification rate **‘on average’**.

Beyond the fact that the inferred mean values of Afrotropical and non-Afrotropical lineages are similar, we highlight broad congruency of speciation rates of both Afrotropical and non-Afrotropical lineages illustrated by the enclosed figure (now presented in the revised version of the supplementary material) resulting from additional BiSSE analyses.

The fact that *within Afrotropical* or *within non-Afrotropical* lineages speciation rates overlap is also consistent with results of BAMM analyses that highlighted the similar diversification regimes of Afrotropical and non-Afrotropical lineages (please see below).

- Another point worth underlining surrounds the results of BAMM analyses in light of the BioGeoBEARS analyses. Three distinct net diversification rate (DR) regimes are inferred under BAMM (please see below, on the left). When visualizing these distinct DR regimes on the results of BioGeoBEARS analyses (please see below, on the right) two interesting conclusions can be made:

First, for the three distinct diversification regimes (labelled 1, 2 and 3) there is a mix of Afrotropical (with nodes circled in red on the tree at the right) and non-Afrotropical lineages (other variously coloured nodes of the same tree). Afrotropical lineages are definitely not associated exclusively with any one diversification regime. Here it is worth mentioning that the lack of Afrotropical lineages for the diversification regime labelled 2 (a subset of the Paspaleae) can be accounted for by the fact that Afrotropical Paspaleae lineages (Afrotropical *Axonopus*, *Hymenachne*, *Ichnanthus* and *Paspalum* spp. plus Afrotropical endemic Paspaleae genera such as *Baptorachis* or *Lecomtella*) are not represented in the phylogeny.

Second, the two shifts in diversification rates (leading to diversification regimes labelled 2 and 3) are not apparently associated with shifts in distribution. The first shift (labelled 2) occurs for a subset of an existing mostly Neotropical clade, and not linked to a change of distribution or biogeographic event (ancestral lineages before and after the DR shift stayed in the Neotropical region). Likewise, the second shift (labelled 3) occurs within a widespread group, and it is not linked to a change of distribution (ancestral lineages before and after the DR shift stayed in the Oriental region).

These new lines of evidence are now included in the revised version of the manuscript (Appendix S1).

3. Finally, in response to the following comment made by reviewer #1:

But I do agree that based on the results presented here as well as the external literature, a decrease of Panicoideae diversity within Africa is very unlikely, so that the conclusions that the hosts and insects diversified in different directions stands.

To address this remark we conducted additional analyses under BiSSE (also presented in the Appendix S1 of the revised version of the manuscript) to infer the pattern of species accumulation through time for both Afrotropical and non-Afrotropical lineages. Based on the most likely BiSSE model, we generated lineages-through-time (LTT) plots using diversity estimates of 900 and 1,034 Afrotropical panicoid species (please see below). Both Afrotropical and non-Afrotropical lineages show increasing patterns of species accumulation, with no distinct up-shifts and more importantly no plateau of diversity (it is also worth mentioning that the slopes would be steeper if we used a complete species-level phylogeny). The slope is higher for non-Afrotropical lineages but this can be attributed to the fact that we used the combined species richness from four continents.

This clear pattern of increasing Afrotropical panicoid diversity therefore provides more support to the ‘*the conclusions that the hosts and insects diversified in different directions*’.

(2) Regarding host specialization. Please acknowledge the possible alternative explanation that the moths could be specialists and yet also be unaffected (in terms of their own diversification) by host diversification, if they remain able to utilize all the descendant hosts of their ancestral hosts. Further, please present as hypothesis rather than fact the idea that an increase in the grass species richness means an increase in potential niches for the moths.

The possible alternative explanation that the moths could be specialists and also be unaffected (in terms of their own diversification) by host diversification is now presented in the main text of the manuscript.

Following your remark we also now present as hypothesis rather than fact the idea that an increase in the grass species richness means an increase in potential niches for the moths.

Depending on the response to these points above and the other reviewer comments, we may be able to evaluate the revision in-house without further need to consult reviewers (particularly Reviewer 1) regarding the remaining issues. Finally, I note that we are willing to editorially overrule Reviewer 3's remaining point about the appropriateness of this particular plant-herbivore group for drawing inferences about grasslands and herbivore evolution. Therefore, this point does not require further comment in the next response to reviewers file.

Thank you for all these clarifications.

Reviewer #1 (Remarks to the Author):

The authors have reworked their manuscript adequately. There are still some imperfections, but I think these are inherent to such analyses of large, complex datasets, and do not in any case affect the validity of the results. I would still like to take the opportunity to reply to some of the authors comments.

We would like to thank the Reviewer #1 for his appreciation of our study.

1) Their BiSSE analyses do show that Afrotropical and non-Afrotropical have the same net diversification rate 'on average'. That said, it does not directly say whether the changes in diversification rates through time identified by the other analyses happened within Africa or not. Again, I do not think that it affects in any way the overall conclusion of the study, but one could imagine based on the analyses presented in this manuscript that the 'increasing rates of speciation over time' for grasses happened outside of Africa. It might very well have concerned both Afrotropical and non-Afrotropical Panicoideae or just Afrotropical Panicoideae, this is just not known. But I do agree that based on the results presented here as well as the external literature, a decrease of Panicoideae diversity within Africa is very unlikely, so that the conclusions that the hosts and insects diversified in different directions stands.

We understand the reviewer's concern.

First, just to provide a definitive answer to the difference in diversification rate between African and non-African grasses, we now provide additional diversification (BiSSE) analyses. These results indicate that speciation rates of both Afrotropical and non-Afrotropical lineages largely overlap as illustrated by the figure presented in the revised version of the supplementary material (see also our response *Ib-* to the Associate Editor). Moreover, we made additional analyses under BiSSE (also presented in the Appendix S1 of the revised version of the manuscript) in order to infer the pattern of species accumulation through time for both Afrotropical and non-Afrotropical lineages. Based on the most likely BiSSE model, we generated lineages-through-time (LTT) plots using either an estimated diversity of 900 and 1,034 Afrotropical panicoid species (see the figure enclosed in our response *Ib-* point 3 to the Associate Editor). The LTTs clearly suggest a pattern of increase in Afrotropical panicoid diversity as corroborative more support for the *'the conclusions that the hosts and insects diversified in different directions'*.

That said, the reviewer's concern may relate more to *'whether the changes in diversification rates through time identified by the other analyses happened within Africa or not.'* The result that speciation rates within Afrotropical and non-Afrotropical lineages are

similar is consistent with the results obtained from BAMM and BioGeoBEARS analyses. BAMM identified two significant changes in diversification rates within Panicoideae (i.e. three regimes of diversification). Importantly, when we combine BAMM results with the BioGeoBEARS inferences, we also estimated that Afrotropical and non-Afrotropical lineages have similar diversification regimes (please see figure enclosed in our response *Ib-* point 2 to the Associate Editor). When we highlight the three distinct regimes of net diversification rates inferred under BAMM, we see a mix of Afrotropical and non-Afrotropical lineages for each shift in diversification rate, suggesting that Afrotropical lineages are not associated with a specific diversification regime or regional distribution. In the case of the Paspaleae, the ancestral lineages before and after the diversification shift remained in the Neotropics. The second shift, within a widespread group of Andropogoneae, is also not linked to a change of distribution.

Overall, the extent of the analyses performed (BAMM, BiSSE and BioGeoBEARS) corroborate our conclusions that (1) diversification is similar between Afrotropical and non-Afrotropical lineages and (2) changes in diversification rates through time are not restricted to a specific area (i.e. continent), and can thus be more readily attributed to trait acquisition or global abiotic events than to ecological opportunity mediated by the geographic colonization of a novel area (continent).

2) Regarding the host-specificity of Sesamiina, I do not find the new analyses extremely compelling. Am I correct that host preference was established by offering 220 species of grasses and sedges to larvae? If so, the full range of host species able to sustain each insect has not been established (only a few portion of the Afrotropical species were evaluated), especially because a number of the hosts were not identified at the species level according to Table S2. I also note from the referenced study (Le Ru et al. 2006) that up to 11 species grew on the same grass species, and out of 36 insects analysed by Le Ru et al., only 21 (the 24 they report was obtained by considering hosts from the same genus as the same host) were found on only one host species. Because the sampling might have missed alternative host plants, it means that at least 40% of the species in this study used to support a 'high level of host specialization' can have multiple host plants. Overall, I do therefore not find strong evidence for a tight association between insects and hosts. I do moreover think that the ancestral state analysis presented in Fig. S5 is not really informative with this respect as it is based on large groups of hosts, which include hundreds to thousands of species. Again, I do not think that this affects the conclusion that the insects and their hosts diversified at different times.

We did not conduct host choice experiments* or 'offer' specific plants to larvae; as described in the main text and in the appendices we collected grasses from habitats suitable for noctuid stemborers (wetlands, marsh of wetlands, riverbanks, forest edges, open grasslands) and that showed traces of infestation by stemborer larvae (generally sesamiine moths but sometime larvae from other groups such as Crambidae or Pyralidae), and reared these larvae in order to get adults. These data collectively represent tens of thousands of plant stems sampled and tentatively identified at the species level (not always possible because diagnostic characters such as the ones associated with the inflorescences were not always present), leading to a total of 220 identified plant species (plus dozen of plants only identified at the subfamily level).

* [host choice experiments would have been irrelevant anyway; larval choice and even larval performance is NOT evidence for what is used in the wild, which can be mediated by abiotic factors, oviposition cues, etc. Performance in the lab does not equate to adaptation or specialization.]

It is also worth underlining that we only collected visually damaged stems of plants although in the field we encountered many more species of grasses, quite a few of which were never infested. Ultimately it leads us to think that we have a reasonable – and unique at this scale – sampling for a group of phytophagous insects.

Overall, after reading the new version of the manuscript and the responses of the authors (and the Le Ru et al. paper suggested by the authors), I think that the lack of association between the diversification of the insects and the grasses is explained by the fact that grass species do not represent distinct ecological niches. First, multiple insect species can prosper on the same grass species. Second, a good fraction of the insects can prosper on multiple grass species (and this fraction is probably underestimated, since not all hosts have been tested). The text added by the authors (lines 398-416) similarly claims that many of the grasses do not represent potential niches for the insects because they are too small for the larvae or because of their high silica content. This therefore explains the disconnect between the diversification patterns of the two groups.

The reviewer's statements are inconsistent with our field observations. In a given locality, though there are usually dozens of potential grass host-plants, but only a subset of them infested, and those usually attacked consistently by particular stemborer species (some localities were sampled several times at different seasons over several years, and many plant species were never found infested).

It is far from being an unusual pattern for endophagous phytophagous insect species, which always exhibit a very high level of host-specialization. Although polyphagous endophages of herbaceous plants are well-known, particularly in the related subtribe Apameina, they tend to be taxonomically restricted and considered exceptional for Sesamiina.

Line 373: 'in this case, C4 grass lineages' → presumably, a grass lineage is not the niche, is it?

To clarify this sentence we corrected it to 'C4 grass species'.

Reviewer #2 (Remarks to the Author):

All my comments on the manuscript have now been addressed, and I congratulate the authors on an impressive body of work and a very interesting contribution to research on diversification!

Sören Nylin

Thank you very much for your second enthusiastic comment on this study.

Reviewer #3 (Remarks to the Author):

The authors propose a revised version of their manuscript that account for the different comments raised by the reviewers. In particular, they added several new analyses to complete and backup several of their results and modify parts of the manuscript to clarify the text.

We thank the Reviewer #3 for his/her second appreciation of our study.

I was happy to see the results of the BiSSE analyses showing that diversification rates of the African panicoids are similar to the whole panicoid clade. This is a comment that I did not raise, but it is an important point to address and the analyses done show nicely this point.

We are glad to read you welcome the BiSSE analyses. In the revised version, we have extended the BiSSE analyses and presented new lines of evidence confirming that rates of speciation are similar between Afrotropical and non- Afrotropical grasses in the Panicoideae.

I also thank the authors for the new divergence time analyses, which shows that the calibration points used are congruent and robust. It does not, however, alleviate the fact that secondary calibrations coming from other studies might not be appropriate if any biased affected these studies, and that it is better control such calibrations by analysing ourselves the available data. This part of the divergence time analyses should however not affect drastically the main results of the manuscript and I am happy with the controls provided here.

We are aware that estimating divergence time analyses are always sensitive to certain uses and priors. We have worked on that in recent years in a number of published empirical studies, providing cautionary tales when one deals with molecular dating (see e.g. Condamine *et al.* 2015 – BMC Evol. Biol.; Toussaint & Condamine 2016 – BJLS; Aghová *et al.* 2018 – MPE). We are always concerned about the molecular dating, especially in insects, and we did our best here to alleviate any methodological issues that can arise due the use of node calibrations. For instance, there is a common practice of using normal prior for secondary calibrations, although it has been shown that such a prior distribution largely biases divergence time estimates (Schenk 2016 – PLoS One). Here we used conservative uniform priors to avoid any biased estimation.

The last point that I have relates to my first comment, which is somewhat also linked with some comments raised by reviewer #1 (e.g. point 8. in minor comments). It is indicated (line 397-398 of the revised manuscript) that the "marked preference [...] for wet habitats may have limited" available niche space for Sesamiina. Similarly, you argue in response to review #1 that Sesamiina have ecological limits on their diversification with potential saturation of the available niche. Recalling your main results, you thus have a large diversification in panicoids, which is not associated with a joint diversification of Sesamiina presumably because of their niche limitation and their preference for wet habitat. It is thus not a counterexample for the role of grasslands in herbivore diversification because the Sesamiina are not adapted to the main conditions found in diversified grasslands, that is dry habitats. To show this, you would need an herbivore group with an ecological niche similar to grasslands that did not follow the same diversification dynamic than grasslands. In contrast, the study nicely show the factors associated with the evolution of this interesting group of insects, but the premise of the manuscript to place their study in the context of the role played by grassland in their evolution is in my opinion flawed. The revised manuscript is not addressing this point and the only way to do it is to change the structure of the introduction and discussion.

As the Associate Editor said, this comment may be overruled but we would still like to address it.

We continue to find the Sesamiina-Panicoideae "system" a fascinating group to investigate insect-plant interactions in a macroevolutionary framework and in the context of grassland evolution. Imperfect or unique though any group may be, the Sesamiina represent a

suite of biological attributes uniquely poised to address questions surrounding the evolution of graminivory in the Old World.

To the best of our knowledge, there are also not many insect groups that harbour such a range of interesting features relevant to such a study: (1) Sesamiina are intimately tied with open grassland habitats; (2) Sesamiina exhibit a very high level of specialization with Panicoideae, one of the most dominant plant groups in the grasslands, such that the two groups have diversified in parallel; (3) being endophagous feeders, it is possible to evaluate their host-range; (4) Sesamiina represent a species-rich group that makes it possible to study at the macroevolutionary scale, i.e. to reliably estimate speciation and extinction rates; and (5) Sesamiina are increasingly well known regarding their higher-level phylogeny, systematics and taxonomy, which is again a difficult standard to reach for many insect groups.

Contrary to the implication of the remarks by reviewer #3, Sesamiina stemborers are adapted to Afrotropical open grasslands, where they constitute one of the most abundant and conspicuous moth groups; though most species exhibit preferences for wet habitats, a non-negligible portion of them (34 species; see Table S2) are also associated with drier conditions; please note that all corresponding information has been added in the revised text and in Table S2, under the *Ecological preferences and distribution* section.

In summary, we have tried to address all the comments and corrections brought up by you and the reviewers, including additional analyses and revisions to the text and presentation. We hope you find our revision appropriate and we are looking forward to receiving your response.

Yours sincerely (on behalf of all co-authors)

Gael J. Kergoat
